# Dietary glucosamine overcomes the defects in αβ-T cell ontogeny caused by the loss of de novo hexosamine biosynthesis

Guy Werlen [1] ✉, Mei-Ling Li[1], Luca Tottone [2,3], Victoria da Silva-Diz[2], Xiaoyang Su[4], Daniel Herranz [2] & Estela Jacinto [1] ✉

T cell development requires the coordinated rearrangement of T cell receptor (TCR) gene segments and the expression of either αβ or γδ TCR. However, whether and how de novo synthesis of nutrients contributes to thymocyte commitment to either lineage remains unclear. Here, we find that T cell-specific deficiency in glutamine:fructose-6-phosphate aminotransferase 1 (GFAT1), the rate-limiting enzyme of the de novo hexosamine biosynthesis pathway (*dn*-HBP), attenuates hexosamine levels, blunts *N*-glycosylation of TCRβ chains, reduces surface expression of key developmental receptors, thus impairing αβ-T cell ontogeny. GFAT1 deficiency triggers defects in *N*-glycans, increases the unfolded protein response, and elevates γδ-T cell numbers despite reducing γδ-TCR diversity. Enhancing TCR expression or PI3K/Akt signaling does not reverse developmental defects. Instead, dietary supplementation with the salvage metabolite, glucosamine, and an α-ketoglutarate analogue partially restores αβ-T cell development in GFAT1[T-/-] mice, while fully rescuing it in ex vivo fetal thymic organ cultures. Thus, *dn*-HBP fulfils, while salvage nutrients partially satisfy, the elevated demand for hexosamines during early T cell development.

T cell development in the thymus is characterized by the expression of distinct T cell receptors (TCR) that function to recognize cognate antigens. While the role of TCR-derived signals in influencing lineage commitment and selection in the thymus is well established[1–3], how metabolism could drive early T cell development remains poorly understood[4]. Given the dependence of developing T cells on the thymic microenvironment for proper ontogeny, it would be important to define how cell fate is influenced by specific nutrients or metabolites that can be acquired from the diet or generated intracellularly by de novo synthesis.

Two major lineages, the αβ- and γδ-T cells, defined by their TCR complex, αβ- and γδTCR respectively, diverge during early thymocyte development (Supplementary Fig. 1a)[1,2]. Rearrangements of the TCR subunit genes, *Tcrd*, *Tcrg*, and *Tcrb* begin at the double negative 2 (DN2; CD4⁻CD8⁻Lin⁻) stage and the separation of αβ- or γδ-T cell lineages is thought to be complete by the DN3 stage[5,6]. This lineage divergence during the DN2 to DN3 stages is coincident with the rearrangement of the TCR loci (*Tcrd*, *Tcrg*, and *Tcrb*). The current model postulates that strong signals propagated by the γδTCR result in higher activation of the ERK pathway and increased induction of transcription factors that eventually promote expression of genes associated with γδ-T cells[7–11]. In contrast, weaker signals skew development towards the αβ-T cell lineage. Whereas αβ-T cells, which undergo extensive thymic selection, exit the thymus in a naïve state

[1]Department of Biochemistry and Molecular Biology, Robert Wood Johnson Medical School, Rutgers, The State Univ. of New Jersey, Piscataway, NJ 08854, USA. [2]Dept. of Pharmacology and Pediatrics, Robert Wood Johnson Medical School, and Rutgers Cancer Institute of New Jersey, Rutgers, The State Univ. of New Jersey, New Brunswick, NJ 08901, USA. [3]Sylvester Comprehensive Cancer Center, University of Miami Miller School of Medicine, FL Miami 33136, USA. [4]Dept. of Medicine, Div. of Endocrinology, Child Health Inst. of New Jersey, Rutgers, The State Univ. of New Jersey, New Brunswick, NJ 08901, USA. ✉e-mail: guy.werlen@rutgers.edu; jacintes@rwjms.rutgers.edu

and acquire effector functions in the periphery, γδ-T cells can become endowed with effector functions in the thymus and subsequently populate the epithelium and mucosa-associated lymphoid tissues[2,12–14]. How metabolism influences lineage fates in the thymus by impacting the expression of cell surface receptors, as well as associated intracellular signals, remains to be further explored.

The expression of membrane proteins, such as the TCR, on the cell surface relies on proper glycosylation, a co/post-translational modification that ultimately contributes to T cell function and homeostasis[15–19]. Glycosylation utilizes hexosamines, which are generated from nutrients including glucose, glutamine, and pyrimidines. The hexosamine biosynthesis pathway (HBP) produces the metabolite uridine diphosphate N-acetylglucosamine (UDP-GlcNAc) that is crucial for glycosylation of proteins, lipids, and small noncoding RNAs[20,21]. UDP-GlcNAc serves as a substrate for glycosylation reactions including the N-linked glycosylation in the endoplasmic reticulum (ER), wherein a high mannose glycan is preassembled and then transferred to consensus Asn-X-Ser/Thr of nascent secretory or membrane-bound polypeptides[22]. It is also used during the remodeling of the N-glycans of glycoproteins undergoing maturation in the Golgi and in O-linked (Ser/Thr) glycosylation of a number of signaling proteins. Defective protein N-glycosylation leads to misfolding, eventually resulting in ER stress[23]. Impaired expression and/or glycosylation of cell surface proteins alters cell signaling and thus influences cell fate. Aberrant lipid glycosylation impacts organelle biogenesis, trafficking, and signaling[22]. Several key intermediates produced by other metabolic/biosynthetic pathways contribute to the HBP. However, the rate-limiting reaction of the de novo synthesis of UDP-GlcNAc is controlled by GFAT1 (glutamine:fructose-6-phosphate amidotransferase 1; also referred to as GFPT1), which directly utilizes glutamine and the glucose metabolite, fructose-6-phosphate (F6P) to generate glucosamine-6-phosphate (GlcN-6-P) (Supplementary Fig. 1b)[24]. UDP-GlcNAc is also generated via salvage pathways, whereby glucosamine (GlcN) or N-acetylglucosamine (GlcNAc) are acquired extracellularly or recycled from intracellular sources and enter the HBP distal to GFAT1-catalyzed reactions (Supplementary Fig. 1b). Dietary supplementation with GlcN has been widely used for anti-inflammatory diseases and has also been reported to have anti-cancer properties[25,26]. However, it is still unclear whether cells use both the de novo and the salvage pathways in synergy or whether one pathway preferentially generates UDP-GlcNAc during a particular developmental stage or immune response.

To address the role of metabolism and the HBP in early T cell development, we generated mice in which Gfat1 was specifically deleted in T cells starting at the DN2 stage of thymocyte maturation. Here, we elucidate how the deletion of GFAT1 results in compromised αβ-T cell development while promoting γδ-T cell expansion. Impaired αβ-T cell ontogeny is characterized by increased TCRβ misprocessing and diminished surface expression of receptors that are required for αβ-T cell development including Notch, CXCR4, and pre-TCR/αβTCR. Despite increased γδ-T cell numbers, γδTCR diversity is attenuated in GFAT1[T−/−] mice. GFAT1 deficiency diminishes UDP-GlcNAc, increases oligomannose-type N-glycans, and increases unfolded protein response (UPR) and the integrated stress response (ISR). The majority of metabolic defects found in GFAT1[T−/−] thymocytes are mimicked by the ablation of rictor albeit to a milder extent. OT-1 TCR overexpression or enhancing PI3K/mTORC2/Akt signaling by the deletion of PTEN do not overcome the developmental defects associated with the absence of GFAT1. Instead, ex vivo supplementation of GFAT1[T−/−] fetal thymic organ cultures (FTOCs) with glucosamine fully rescues αβ-T cell ontogeny. Interestingly, dietary supplementation of GFAT1[T−/−] mice with this salvage metabolite enhances the survival of DN and SP cells but does not restore the developmental block during the highly proliferative DN3-DN4 and CD8-ISP stages. Our findings unravel the importance of the de novo HBP to meet the increased demand for hexosamines during T cell development in the thymus.

## Results

### αβ-T cell developmental defects occur in the absence of GFAT1

To analyze the role of GFAT1 during T cell ontogeny (Supplementary Fig. 1a), we used the proximal Lck-Cre promoter to ablate Gfpt1 (herein referred to as Gfat1) at the double negative (CD4−CD8−; DN) 2 (DN2) stage and thereafter. Analysis of protein expression confirmed that GFAT1 was specifically deleted in thymocytes of GFAT1[T−/−] mice, but not lung or liver cells (Fig. 1a). Compared to GFAT1[fl/fl] (wild-type; WT) or GFAT1[+/+] Lck-Cre[+/−] (Lck-Cre[T+/−]) control WT animals, GFAT1[fl/fl] Lck-Cre[+/−] (GFAT1[T−/−]) mice had a 4-5 fold decreased number of thymocytes (Fig. 1b), leading to a similar loss of peripheral T cells in the spleen (Fig. 1c, d). While no significant differences in CD4 vs CD8 repartition were observed in GFAT1[T−/−] splenic peripheral T cells as compared to WT or Lck-Cre[T+/−] (Fig. 1e), total cell numbers of CD4+ T helper (CD4+ $T_H$) and CD8+ T cytotoxic (CD8+ $T_C$) were dramatically decreased in the spleen of GFAT1[T−/−] mice (Fig. 1f). About a 50% loss of double positive (CD4+CD8+ DP) and CD4-single positive (CD4-SP) thymocytes in the absence of GFAT1 accounts in part for the decreased peripheral CD4+ $T_H$ (Fig. 1g). In contrast, the proportion of CD8+ cells were augmented by ~50% in the thymus of GFAT1[T−/−] mice (Fig. 1g). However, this robust increase did not translate into a higher cell number of those subsets (Fig. 1h). Instead, the number of CD8+ thymocytes, as well as both DP and CD4-SP subsets, dropped dramatically, reflecting the tremendous loss in total thymocyte as well as peripheral T cell numbers observed in GFAT1[T−/−] mice (Fig. 1b, d, f). The augmented percentage of CD8+CD4− subsets could be due to a developmental defect at the CD8-immature single positive (CD8-ISP) stage. CD8-ISPs are a rapidly dividing transitional subset that rearranges TCRα-chains prior to the DP stage wherein a functional αβTCR is expressed together with both CD8 and CD4 coreceptors. CD8-ISP cells express high amounts of CD8 and CD147 on their surface, but low amounts of pre-TCR/TCR[27–29]. In WT and Lck-Cre[T+/−] mice, CD8-ISP represents the minority population of CD8+ thymocytes (~30%) while mature CD8-SP constitutes the majority (~70%) (Fig. 1i). In stark contrast, >85% of CD8+ cells from GFAT1[T−/−] mice were CD8-ISP and <10% were mature CD8-SP (Fig. 1i), indicating an accumulation of the former subset. Nevertheless, the total numbers of CD8-ISP thymocytes lacking GFAT1 remained significantly lower than their WT or Lck-Cre[T+/−] counterparts (Fig. 1j), which probably accounts in part for the profound loss of DP cells in GFAT1[T−/−] mice (Fig. 1h). Overall, our results indicate pronounced defects in αβ-thymocyte maturation in the absence of GFAT1.

### Defective expression of key surface receptors at the DN stage occurs in GFAT1-deficient thymocytes

Since there was a profound increase in the proportion of DN thymocytes in the absence of GFAT1 (Fig. 1g), we next characterized the different DN subsets to obtain clues on the role of GFAT1 in early T cell development. Using the cell surface markers, CD44 and CD25, we checked for defects along the DN differentiation stages. A slight decrease in relative and absolute numbers of DN1 (CD44+CD25−) cells occurred in GFAT1[T−/−] mice, while DN3 (CD44−CD25+) numbers were dramatically increased by 2.5-fold (Fig. 2a, b). A 2-fold decrease in DN4 (CD44−CD25−) subsets suggests a differentiation defect prior to the DN4 stage that is consistent with DN3 cell accumulation. The latter could account for the higher proportion of DN thymocytes observed in GFAT1[T−/−] mice (Fig. 1g). CD27-expressing DN3b thymocytes differentiate from early DN3a (Lin−CD25+CD44−CD27[low]) cells upon Notch1 signaling and successful rearrangement of TCRβ-chains[30](Supplementary Fig. 2a). GFAT1-deficient DN3a and DN3b cells had significantly decreased levels of CD27, Notch1 and CXCR4 on their surface (Fig. 2c, d and Supplementary Fig. 2b). This could in part contribute to flawed maturation to the DN3b and subsequently DN4 stages, as accounted for by suboptimal proportions of Notch1+TCRβ+DN3b and Notch1−TCRβ+DN4 subsets, respectively (Fig. 2e). Notch1 and the chemokine receptor, CXCR4 are essential for the commitment of developing thymocytes towards the

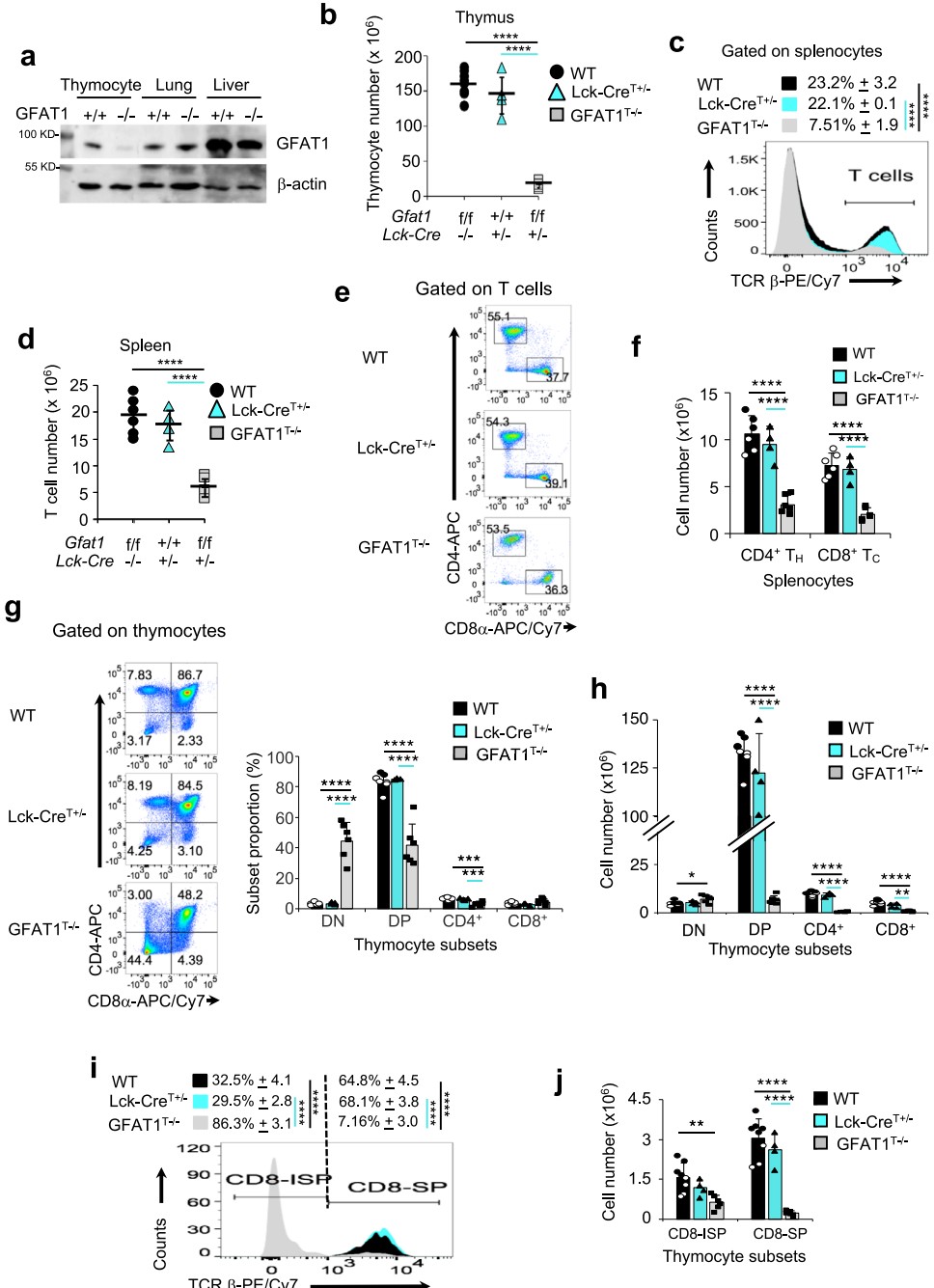

**Fig. 1 | Defective αβ-T cell development in the absence of GFAT1. a** Total cell extracts from thymocytes, lung, and liver from male and female WT (+/+) or mice with specific deletion of GFAT1 in T cells (GFAT1^(T−/−)) were subjected to SDS-PAGE and immunoblotted using the indicated antibodies. Representative blot out of two independent experiments with similar results is shown. **b** Thymocytes from male and female 5-week-old *Lck-Cre^(−/−)/Gfat1*^(f/f) (WT), *Gfat1*^(+/+)/*Lck-Cre*^(+/−) (Lck-Cre^(+/−)) or *Lck-Cre*^(+/−)/*Gfat1*^(f/f) (GFAT1^(T−/−)) littermates or age-matched were harvested and counted by trypan blue exclusion. Each symbol represents one mouse (*n* = 4–8 mice each). **c–f** Splenocytes of 5-week-old male and female *Lck-Cre^(−/−)/Gfat1*^(f/f) (WT), *Gfat1*^(+/+)/*Lck-Cre*^(+/−) (Lck-Cre^(+/−)) or *Lck-Cre*^(+/−)/*Gfat1*^(f/f) (GFAT1^(T−/−)) littermates or age-matched were stained for TCRβ, CD4 and CD8α followed by flow cytometric analysis. Shown are the proportions of T cells among splenocytes (**c**); total number of T cells in each mouse (**d**) (*n* = 4–6 mice each); proportions of CD4 T helper (CD4⁺ T_H) and CD8 T cytotoxic (CD8⁺ T_C) cells among peripheral T cells (**e**); total cell number of CD4⁺ T_H

and CD8⁺ T_C subsets in the spleen of the respective mice (**f**) (*n* = 4–6 mice). Representative FACS plots (**c, e**) from three experiments with similar results are shown. **g–j** Thymocytes from male and female *Lck-Cre^(−/−)/Gfat1*^(f/f) (WT), *Lck-Cre*^(+/−) (Lck-Cre^(+/−)) or *Lck-Cre*^(+/−)/*Gfat1*^(f/f) (GFAT1^(T−/−)) littermates or age-matched were stained for CD4, CD8α, CD147 & TCRβ followed by flow cytometric analysis to measure the proportion of thymocyte subsets as indicated in each quadrant. Bar graph represents proportion (**g**) or absolute cell number (**h**) (*n* = 4–8 mice). CD8⁺CD4⁻ thymocytes were gated for TCRβ^(low)/CD147⁺ to distinguish CD8-ISP (TCRβ^(low)) from lineage-committed CD8-SP (TCRβ^(high)) cells (**i**). Bar graph (**j**) represents absolute cell numbers of CD8-ISP and CD8-SP (*n* = 4–8 mice). Representative FACS plots (**g, i**) from three experiments with similar results are shown. All data from (**b–d, f–j**) are mean ± SD. *p < 0.05, **p < 0.01, ***p < 0.001, ****p < 0.0001 using one-way ANOVA followed by Tukey's post-hoc test. Source data are available for (**a–j**).

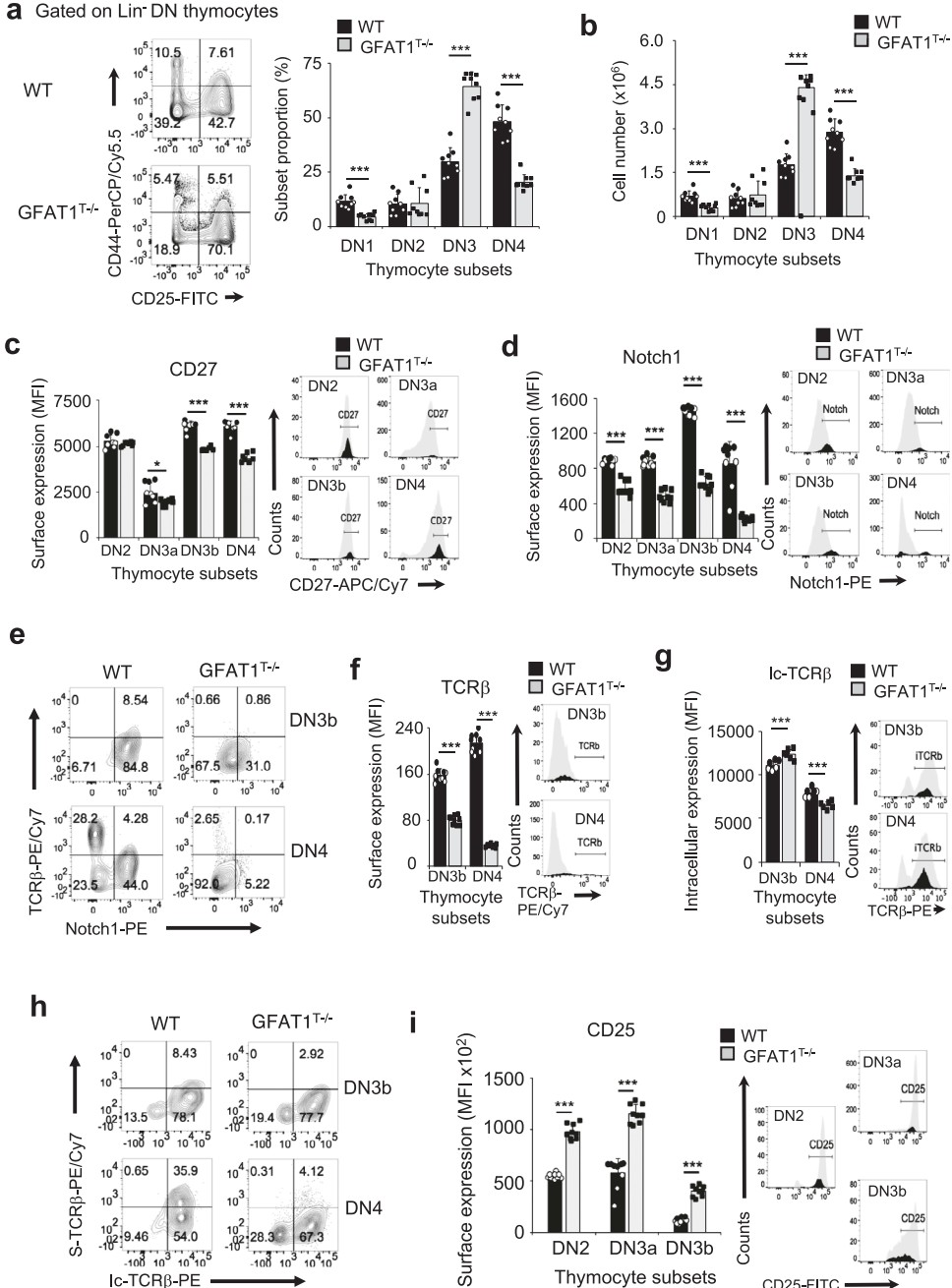

**Fig. 2 | Defective expression of key surface receptors at the DN stage in GFAT1-deficient thymocytes.** Thymocytes from male and female WT (*Lck-Cre$^{-/-}$/Gfat1$^{f/f}$*) and knockout (*Lck-Cre$^{+/-}$/Gfat1$^{f/f}$*; referred to as GFAT1$^{T-/-}$) littermates were stained for Lin, CD4, CD8α, CD25, CD44, CD27, Notch1, and TCRβ followed by flow cytometric analysis to measure the proportion of Lin$^-$ DN subsets as indicated in each quadrant (a-f, i) or further stained for intracellular (ic) TCRβ (g-h) before flow cytometry. **a**, **b** FACS plots and bar graphs represent proportion (**a**) or absolute cell number (**b**), (*n* = 9 mice). **c**, **d** DN3 thymocytes were further analyzed for the proportion of CD27$^-$DN3a and CD27$^+$DN3b subsets (see also Supplementary Fig. 2a) followed by analysis of CD27 and Notch1 surface expression. Bar graphs of median fluorescence intensity (MFI) of CD27 (**c**) and Notch1 (**d**) expressed on the surface of DN2, DN3a, DN3b, and DN4 thymocytes (*n* = 3 mice with 3 technical repeats) and respective representative FACS plots are shown. **e**–**g** Representative FACS plots

showing DN3b and DN4 subsets that express Notch1 and TCRβ is shown (**e**). Percentage is indicated in each quadrant. Representative FACS plots and bar graphs represent median fluorescence intensity (MFI) of TCRβ expression on the surface of DN3b and DN4 cells (**f**)(*n* = 9 mice) or ic-TCRβ expression (**g**)(*n* = 6 mice). **h** DN3b and DN4 subsets were analyzed for ic- and surface (s) TCRβ staining. Representative FACS plots with the percentage of cells in each quadrant are shown. **i** CD25 expression on the surface of DN2, DN3a, and DN3b thymocyte subsets. Bar graph of median fluorescence intensity (MFI) and respective representative FACS plots of CD25 staining are shown (*n* = 9 mice). FACS plots from (**a**–**i**) are representative of at least 3 experiments with similar results. All data (**a**–**d**, **f**, **g**, **i**) are mean ± .SD. *$p < 0.05$, **$p < 0.01$, ***$p < 0.001$ using a two-sided Student's *t* test. Source data are available for (**a**–**i**).

αβ-T cell lineage[31,32]. It is noteworthy that the amount of Notch1 and CXCR4 on the surface of WT DN thymocytes almost doubled from the DN2/DN3a to the DN3b stage, while it essentially remained unchanged along those developmental stages in the absence of GFAT1 (Fig. 2d and Supplementary Fig. 2b). Similarly, while surface expression of pre-TCR as

measured by TCRβ median fluorescence intensity (MFI), gradually increased on DN3b to DN4 WT cells, its amount greatly decreased in GFAT1$^{T-/-}$ cells (Fig. 2f). Interestingly, measurement of newly synthesized intracellular TCRβ chains showed higher amounts of this receptor in GFAT1-deficient DN3b cells as compared to WT counterparts (Fig. 2g),

suggesting that misprocessing of, or a defect in shuttling TCRβ chains through the Golgi, could account in part for lower pre-TCR surface expression. Although a similar percentage of WT and GFAT1$^{T-/-}$ DN3b thymocytes expressed intracellular TCRβ, ~3x less cells expressed this receptor on their surface in the absence of GFAT1 (2.92 vs 8.43%) (Fig. 2h). The proportion of GFAT1-deficient DN4 cells expressing both intracellular and surface TCRβ was ~9x lower than WT cells (4.12 vs 35.9%). Altogether, our findings reveal that GFAT1 is required for proper αβ-T cell development and that its absence markedly affects the DN3 to DN4 as well as CD8-ISP to DP transitions, which correspond to the TCR β- and α-chain selection stages, respectively (Supplementary Fig. 1a). The defective αβ-T cell development in the absence of GFAT1 is accompanied by suboptimal expression of CD27, Notch1, CXCR4 and the pre-TCR β-chain at the DN3b stage.

We then further examined the expression of the αβTCR in different thymic subsets. Overall, there was a dramatic decrease in TCRβ expression on the surface of GFAT1$^{T-/-}$ thymocytes at every stage of T cell ontogeny that is known to express pre-TCR or αβTCR (Supplementary Fig. 2c), as well as on mature GFAT1$^{T-/-}$ αβ-T cells in the periphery (Supplementary Fig. 2d). This was accompanied by diminished TCRβ mRNA levels in GFAT1-deficient thymocytes (Supplementary Fig. 2e). Reduced αβTCR expression resulted in its compromised signaling as revealed by lower levels of CD5 on the surface of GFAT1-deficient thymocytes and peripheral T cells (Supplementary Fig. 2f, g). CD5 expression normally correlates to the strength of TCR-signaling[33]. While GFAT1 deficiency induced a significant decrease in pre-TCR/αβTCR expression on the surface of CD8-ISP thymocytes (Supplementary Fig. 2c, inset), it profoundly increased CD147 and CD98 surface expression (Supplementary Fig. 2h). CD147 facilitates the surface expression of the lactate transporter, while CD98, in part, controls the transport of large neutral amino acids[34,35]. Thus, our results imply that CD8-ISP thymocytes lacking GFAT1 have a distinct metabolism as compared to WT counterparts. Similarly, CD127 (IL-7Rα), which is a critical cytokine receptor for DN thymocyte homeostasis[36] and CD25 (IL-2Rα), were significantly increased on the surface of GFAT1-deficient DN thymocytes (Fig. 2i and Supplementary Fig. 2i). Notably, the enhanced CD25 surface expression coincided with a dramatic increase of its mRNA levels in the absence of GFAT1 (Supplementary Fig. 2j), suggesting the existence of a feedback mechanism that upregulates the expression of essential cytokine receptors in the absence of a functional dn-HBP. Overall, our findings reveal that GFAT1 controls the expression of surface receptors relevant to the metabolism and lineage commitment of developing αβ-T cells. Our findings also suggest that GFAT1 is essential for the optimal processing of cell surface receptors such as the TCR.

## Defective hexosamine and nucleotide biosynthesis, glycosylation, and increased unfolded protein response occur in the absence of GFAT1

Given the defects in surface expression of receptors critical for αβ-T cell development, we next analyzed by liquid chromatography-mass spectrometry (LC-MS) how the absence of GFAT1 affects the HBP and the production of other cellular metabolites. Notably, UDP-GlcNAc was significantly diminished in the absence of GFAT1 (Fig. 3a), which could explain in part the defective processing and surface expression of highly N-glycosylated receptors such as the αβTCR. To further investigate if GFAT1 deficiency affected N-glycosylation of glycoproteins, we conducted N-glycomics analysis by mass spectrometry. To facilitate the recovery of sufficient thymocytes in the absence of GFAT1, we used WT and GFAT1$^{T-/-}$ mice under the OT-1 TCR transgenic background. We found that there was increased proportion of the oligomannose/paucimannose N-glycans, while complex/hybrid types were decreased in the absence of GFAT1 (Supplementary Fig. 3a). To further substantiate defective N-glycosylation in the absence of GFAT1, we analyzed the expression of TCRβ in whole cell lysates of thymocytes treated with or

without the proteasome inhibitor MG132. The reduced expression of TCRβ caused by the loss of GFAT1 was increased to WT levels following MG132 treatment (Fig. 3b), indicating augmented protein degradation upon GFAT1 deficiency. Furthermore, lectin pull-down assay showed increased binding of TCRβ to Galanthus nivalis lectin (GNL) during MG132 treatment of OT-1/GFAT1$^{T-/-}$ but not OT-1/WT thymocytes. In contrast, the proteasome inhibitor did not alter the amount of TCRβ pulled-down by the lectin SNA (Sambucus nigra agglutinin) from OT-1/WT or OT-1/GFAT1-deficient thymocyte extracts. Since GNL binds to α-1,3-mannose that are exposed in the early steps of N-glycan remodeling, while SNA binds to terminal sialic acid residues, our results suggest that early glycan misprocessing accounts in part for the increased degradation of TCRβ in the absence of GFAT1. In addition to defective N-glycosylation, total O-GlcNAcylation was also diminished in GFAT1-deficient αβ-thymocytes (Fig. 3c). Collectively, these findings reveal that in the absence of dn-HBP, there is limited UDP-GlcNAc, consequently leading to defective N-glycosylation and O-GlcNAcylation of glycoproteins.

In contrast to UDP-GlcNAc, the levels of glutamine and glucose, which are both required substrates of the HBP, were augmented in GFAT1$^{T-/-}$ thymocytes (Fig. 3a). This would be coherent with elevated surface expression of CD98 and GLUT1, respectively, on these cells (Fig. 3d, e). Reduced expression of glutaminase (GLS) in GFAT1$^{T-/-}$ thymocytes (Fig. 3f) could also contribute to glutamine build-up and consequently glutamate decrease (Fig. 3a). UTP, which also enters the HBP, was profoundly reduced in GFAT1$^{T-/-}$ thymocytes (Fig. 3a). It is notable that other nucleotides such as ATP, CTP, GTP, as well as pyrimidines and purines were also downregulated in GFAT1-deficient thymocytes, suggesting a broad defect in nucleotide metabolism in those cells (Fig. 3a). Thus, we examined the expression of CAD (carbamoyl-phosphate synthetase 2, aspartate transcarbamylase and dihydroorotase), the key metabolic enzyme involved in de novo synthesis of pyrimidines. The absence of GFAT1 abolished CAD protein expression altogether (Fig. 3f), suggesting a post-transcriptional reduction of CAD expression. Like glutamine, Arg, Gly, Pro, Tyr, Ser as well as all the essential amino acids were increased in the absence of GFAT1 (Fig. 3a and Supplementary Fig. 3b). In contrast, there was a significant decrease in the other non-essential amino acids (NEAA) Glu, Asp, Asn and Ala (Fig. 3a).

We previously reported that mTORC2 modulates GFAT1 and the biosynthesis of hexosamines[37,38]. Surprisingly, the absence of GFAT1 also markedly diminished the expression of rictor (Fig. 3f and Supplementary Fig. 3c), while modestly reducing mTOR suggesting that mTORC2 is also somewhat compromised in GFAT1$^{T-/-}$ thymocytes. However, in the absence of GFAT1, basal mTORC2 activity as indicated by the phosphorylation of Akt at Ser473 was fairly similar to WT level (Fig. 3f). In contrast, rictor ablation had no effect on GFAT1 expression but slightly reduced mTOR expression and Ser473 Akt phosphorylation (Fig. 3f). As documented in the literature, rictor deficiency affects the processing and surface expression of receptors relevant for T cell ontogeny[27,39,40], albeit less dramatic as compared to the loss of GFAT1. We therefore compared the levels of metabolites in the absence of rictor with GFAT1-deficiency. Metabolites in rictor$^{T-/-}$ thymocytes went on a similar trend as that found in GFAT1-deficient cells, although in most cases, less drastic (Fig. 3a). Notably, UDP-GlcNAc, GlcNAc, purine and pyrimidine biosynthesis metabolites were generally decreased in the absence of rictor. Together, these findings reveal that the absence of GFAT1 diminishes UDP-GlcNAc, NEAA, purines, pyrimidines and nucleotides in developing thymocytes while the absence of rictor has relatively less severe alterations in overall metabolites.

Next, we employed quantitative proteomics to determine broad effects of the absence of the dn-HBP vs the loss of rictor (Fig. 3g and Supplementary Fig. 3d). Abrogating GFAT1 had generally more dramatic effects on the expression of proteins involved in metabolic and

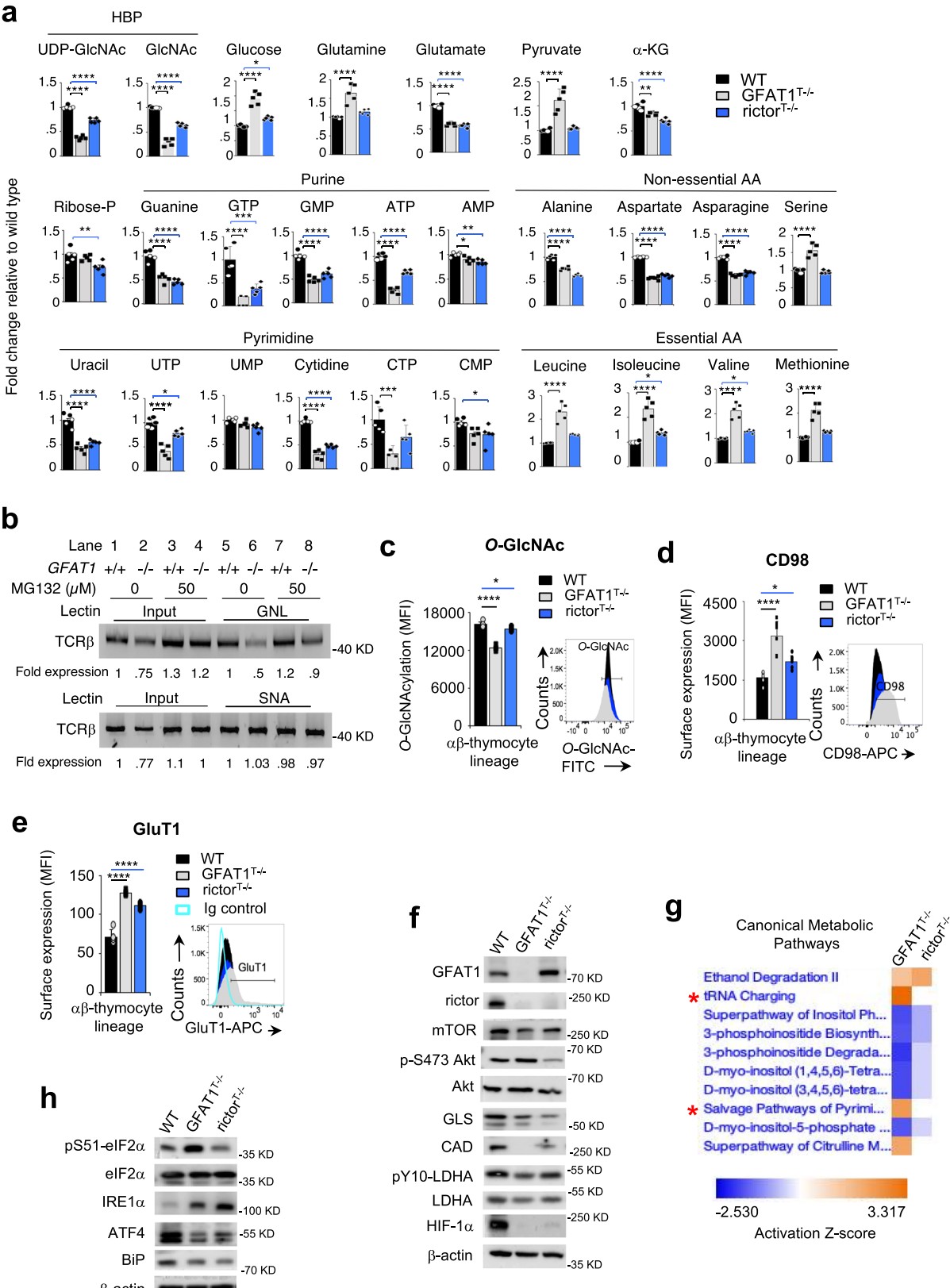

signaling pathways (Fig. 3g and Supplementary Fig. 3d). Consistent with defects in CAD, our proteomics analysis revealed increased expression of proteins involved in the pyrimidine salvage pathway (Fig. 3g). In GFAT1[T−/−] thymocytes, there was also increased expression of proteins involved in oxidative phosphorylation concomitant with aberrant glycolysis as indicated by decreased phosphorylation of

lactate dehydrogenase A (LDHA) and abolished expression of HIF-1α (Fig. 3f).

Our proteomics data also revealed augmented expression of components of the unfolded protein response (UPR) in the absence of GFAT1 (Supplementary Fig. 3d), which would be consistent with diminished UDP-GlcNAc and increased protein degradation due to

**Fig. 3 | Defects in hexosamine, glycosylation, and increased unfolded protein response in GFAT1-deficient thymocytes.** Thymocytes from male and female age-matched WT, GFAT1$^{T-/-}$ and /or rictor$^{-/-}$ mice were used for the following except for b. **a** Metabolites were extracted and analyzed from equivalent thymocyte numbers. Graphs represent mean fold changes relative to WT ($n = 5$ independent samples; [1 WT, pool of 3-5 GFAT1$^{T-/-}$ and 2–4 rictor$^{-/-}$ per sample). Error bars denote SD. *$p < 0.05$, **$p < 0.01$, ***$p < 0.001$, ****$p < 0.0001$ using one-way ANOVA followed by Tukey's post-hoc test. See also Supplementary Fig. 3b. **b** Thymocytes were harvested from male and female OT-1/WT and OT-1/GFAT1$^{T-/-}$ mice and treated with vehicle or MG132 for 4 h. Lysates were subjected to lectin pull-down assays followed by immunoblotting of TCRβ. Fold changes relative to TCRβ expression in untreated OT-1/WT cells (lane 1 for input, lane 5 for pull down) are indicated below each blot. Representative blots from two independent experiments with similar results are shown. **c–e** Thymocytes were harvested, intracellularly stained (**c**) or surface stained (**d**, **e**), then analyzed by flow cytometry. Bar graphs represent mean fluorescence intensity (MFI). $n = 5$–8 mice for (**c–e**) and the respective representative FACS plots from three experiments with similar results are shown. All data are mean ± SD. *$p < 0.05$, **$p < 0.01$, ***$p < 0.001$, ****$p < 0.0001$ using one-way ANOVA followed by Tukey's post-hoc test. **f, h** Protein extracts of thymocytes were resolved and subjected to immunoblotting. β-actin was used as loading control for all blots. Shown are representative immunoblots from at least three independent experiments with similar results. Closest MW (KD) marker is indicated for each blot. **g** Protein extracts from thymocytes were subjected to quantitative proteomics and data were analyzed by Ingenuity Pathway Analysis (IPA). Shown is the heat map of statistically significant ($p < 0.05$ as determined using two-sided Student's $t$ test) canonical metabolic pathway alterations in GFAT1- and rictor-deficient thymocytes relative to WT. Red asterisks denote an increase of both tRNA charging and salvage pathways of pyrimidine. Pathways are ranked according to the z-score that predicts upregulation (orange) or downregulation (blue). See complete heat map in Supplementary Fig. 3d. Source data are available for a-f,h.

mis-glycosylation. Corroborating these data, we found increased phosphorylation of eIF2α, which is involved in the integrated stress response (ISR) (including UPR) (Fig. 3h). Consistent with ISR[41], GFAT1-deficient thymocytes also had enhanced expression of proteins involved in tRNA charging (Fig. 3g). However, not all UPR proteins had augmented activity or expression. While the expression of IRE1α was also increased in GFAT1$^{T-/-}$ and rictor$^{T-/-}$ thymocytes, the amounts of Bip and ATF4 were decreased in the same cells (Fig. 3h). Altogether, our results reveal that the absence of GFAT1 is not only linked to defective hexosamine biosynthesis (HB) but also to compromised nucleotide metabolism and oxidative phosphorylation as well as to increased ISR and UPR.

## GFAT1 deficiency decreases the viability and proliferation of αβ-lineage thymocytes but increases γδ-thymocyte cell numbers

Since the ontogeny of γδ-T cells also commences in the thymus, we examined whether GFAT1 is required for their maturation. The analysis of γδ vs αβ lineage-committed thymocytes revealed a dramatic 20-fold increase in the percentage of GFAT1-deficient γδ-T cells, while commitment to the αβ-lineage was down ~2.5 times in GFAT1$^{T-/-}$ mice (Fig. 4a). Albeit less striking, the total number of γδ-thymocytes was also significantly augmented in the absence of GFAT1 (Fig. 4b). In GFAT1$^{T-/-}$ mice, the proportions of the two T cell lineages were skewed in favor of mature γδ-T cells in the thymus (Supplementary Fig. 4a). γδ-T cell proportion also doubled in the spleen of GFAT1$^{T-/-}$ mice although αβ-T cells dominated in this organ (Supplementary Fig. 4b).

Since αβ- and γδ-T cell lineages diverge at the DN3 stage of thymocyte development, a change in maturation/commitment could account for the augmented γδ-cells observed in the absence of GFAT1. During the first 9 weeks after birth, the percentage of WT DN cells expressing γδTCR decreased sharply from 15% after birth to ~7% after 1 week, while the TCRβ expressing subset exhibited the opposite fate, abruptly increasing from <5% to >20% after 1 week and remaining above 20% thereafter (Fig. 4c). In contrast, the percentage of GFAT1-deficient γδ-DN thymocytes gradually increased to 3x more than the proportion found in WT by 9 weeks. As for GFAT1-deficient DN subsets expressing TCRβ (pre-TCR) on their surface, it remained below 5% from newborn to 9 weeks old (Fig. 4c), suggesting that GFAT1 is specifically required for the development of αβ- but may be dispensable for γδ-T cell ontogeny. To further interrogate whether GFAT1 has a role in γδ-T cell maturation, we analyzed γδ-T cell subsets. Whereas immature γδ-thymocytes are CD24$^+$CD73$^-$, downregulation of CD24 and upregulation of CD73 underscores the maturation of γδ-T cells[11,42]. While GFAT1-deficient fetal/neonatal mice had slightly higher proportions of CD24$^+$CD73$^-$ immature, CD24$^+$CD73$^+$ "maturing" and CD24$^-$CD73$^+$ mature γδ-cells compared to their WT counterparts, only the latter subset remained increased in 7-week old GFAT1$^{T-/-}$ mice (Fig. 4d). γδ-thymocytes are generated in waves of cells expressing TCR with similar variable domains (V). Thus, we stained WT and GFAT1-deficient γδ-thymocytes with commercially available antibodies recognizing the distinct V domains of the TCRγ-chain, Vγ1.1, Vγ2, and Vγ3 (according to the Garman nomenclature). Vγ3 is mainly expressed in fetal/neonates, while the vast majority of γδ-cells in adult mice expressed Vγ1.1 and to a lesser extent Vγ2[43]. Compared to WT fetal/neonates (0 w), Vγ3 was prominently expressed on the surface of >50% of CD24$^-$CD73$^+$ mature γδ-thymocytes from GFAT1$^{T-/-}$ mice (Fig. 4e). While Vγ2 and Vγ1.1 were already well established in WT counterparts, they were barely discernible among γδ-cells from GFAT1-deficient fetal/neonates (Fig. 4e). In sharp contrast to WT adult mice (7 w) in which ~70% of γδ-thymocytes expressed Vγ1.1, <10% of GFAT1-deficient counterparts bore this particular Vγ domain (Fig. 4e). The proportion of Vγ2 and Vγ3 was even lower (2.3% and 1.4%, respectively). The identity of the Vγ domain that was expressed on the majority of γδ-thymocytes lacking GFAT1 in 7-week-old mice remains to be identified. Similar trends of expression of distinct Vγ domains were found when total γδ-thymocyte numbers were analyzed (Fig. 4f). Together, these findings suggest that GFAT1 plays a role in generating γδ-TCR diversity.

We next examined whether a proliferation defect could account for the altered ratio of αβ- vs γδ-DN thymocytes in the absence of GFAT1. Upon staining with CFSE and gating for DN4 thymocytes expressing TCRβ, we found markedly reduced basal proliferation of GFAT1-deficient cells by 24 h (~20X less) and 48 h (~15X less) (Fig. 4g upper panel). Fairly similar results were observed when gating for the highly proliferating CD8-ISP or the slightly less dividing DP thymocytes (Supplementary Fig. 4c–f). Although the survival ratio of GFAT1$^{T-/-}$ cells cultured ex vivo over 48 h was overall similar to WT counterparts (Fig. 4h), only ~1/2 as many living TCRβ$^+$-DN4 cells could be recovered from GFAT1$^{T-/-}$ mice (Fig. 4h start of culture). These findings indicate that GFAT1 is essential for the survival and/or proliferation of αβ-lineage cells in the thymic environment in vivo.

When we compared WT DN thymocytes expressing γδTCR with pre-TCR$^+$-DN4 cells, we found that the former proliferated less. However, there was a ~4-fold increased proportion of robustly dividing γδ-DN cells as compared to their DN4 counterparts (~4% γδ DN vs 1% DN4). Abrogating GFAT1 further augments the percentage of these highly dividing γδ-DN cells reaching 9% after 24 h and ~16% after 48 h compared to a steady 4% of WT counterparts (Fig. 4g lower panel). Hence, although overall proliferation of GFAT1-deficient γδ-DN was lower compared to WT (Fig. 4g lower panel), the increased proportion of rapidly expanding subsets could contribute to the augmented thymic γδ-T cell number in GFAT1$^{T-/-}$ mice (Fig. 4b). Furthermore, in contrast to the TCRβ$^+$-DN4 subset, GFAT1-deficient γδ-DN cells have fairly similar viability at the start of culture as their WT counterparts and the absence of GFAT1 became only relevant after 24 h of culture (Fig. 4i). These findings reveal that GFAT1 is not essential for γδ-T cell proliferation.

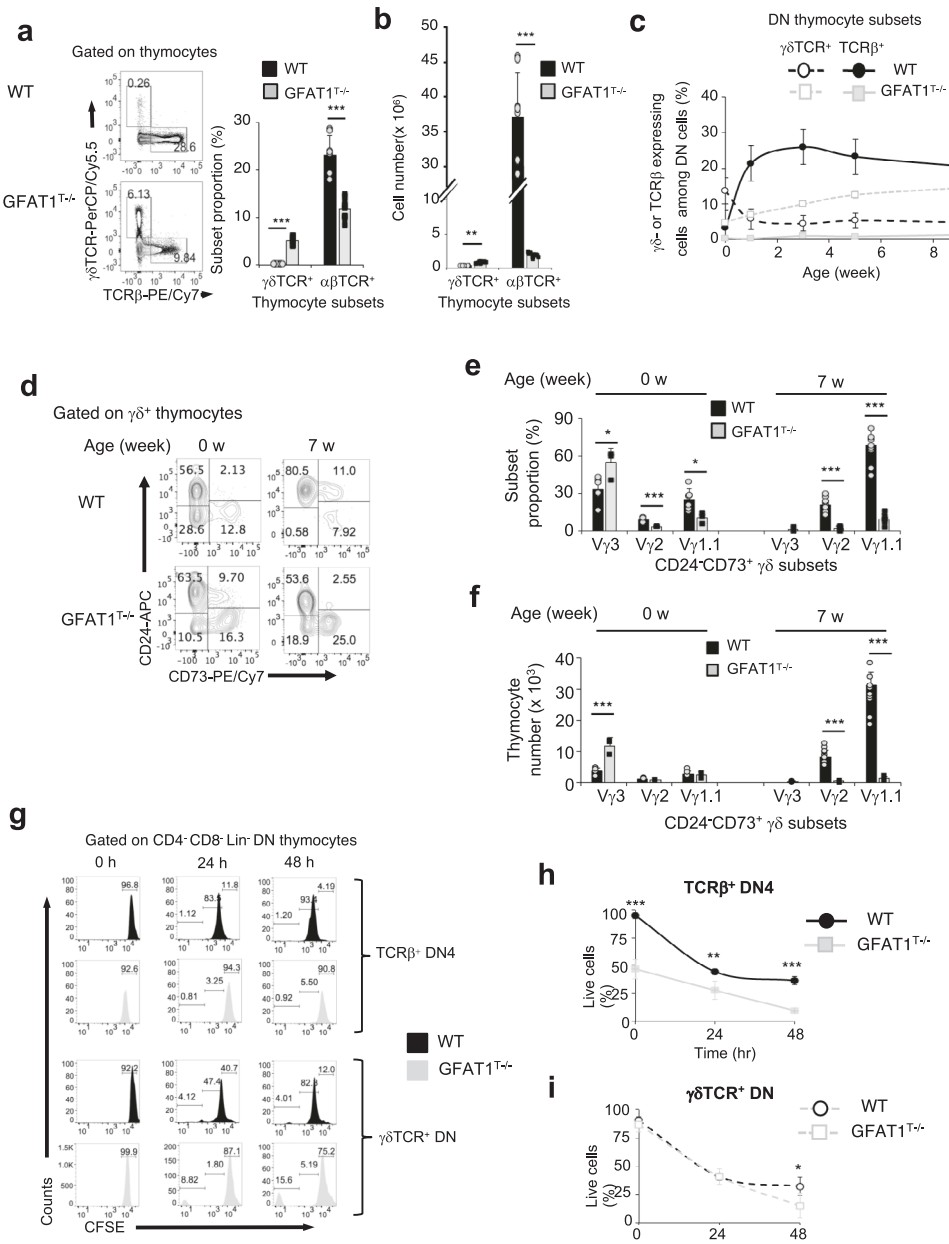

**Fig. 4 | GFAT1 deficiency decreases the viability and proliferation of αβ-thymocytes but increases γδ-thymocyte cell numbers.** Thymocytes from male and female age-matched WT and GFAT1$^{T-/-}$ mice were used for the following experiments. **a-b** Thymocytes were stained for CD4, CD8, CD44, CD25, TCRβ, γδTCR and analyzed by flow cytometry. Representative plots with bar graph of the proportion of subsets (**a**) or absolute cell numbers (**b**) expressing surface TCRβ (αβ-lineage) or γδTCR (γδ-lineage) are shown (**a, b**, $n = 8$–12 mice). **c** 0-, 1-, 3-, 5- and 9-week mice were stained for Lin, CD4, CD8α, CD25, CD44, TCRβ and γδTCR followed by flow cytometric analysis of αβ- (express TCRβ) or γδ-lineage subsets among DN thymocytes. The percentage of each lineage is plotted relative to the mouse age ($n = 1$–6 mice). **d**–**f** Thymocytes from e18/19 and 7-week-old mice were harvested and stained with γδTCR, CD24, CD73, Vγ1.1, Vγ2 and Vγ3 followed by flow cytometric analysis. Partition analysis of immature (CD24$^-$CD73$^-$ & CD24$^+$CD73$^-$), maturing (CD24$^+$CD73$^+$) and mature (CD24$^-$CD73$^+$) γδ-thymocyte subsets (**d**).

Shown is a representative plot of three experiments with similar results. Percentage is indicated in each quadrant. Bar graphs represent proportion (**e**) and total number (**f**) of mature CD24$^-$CD73$^+$γδTCR$^+$ γδ-thymocytes expressing either Vγ1.1, Vγ2 or Vγ3 (**e, f**, $n = 3$–9 mice). **g** Thymocytes from 5-week-old mice were labeled with CFSE and cultured ex vivo for the indicated hours then harvested and stained as in (**c**), followed by flow cytometry. Gating was set on live thymocytes and either TCRβ$^+$-DN4 (upper panel) or γδTCR$^+$ DN cells (lower panel). Shown is one representative experiment out of three with similar results. **h** Thymocytes were cultured ex vivo for 24 h or 48 h and stained for Lin, CD4, CD8α, CD25, CD44, Annexin V, and TCRβ. Graph shows viability of TCRβ$^+$-DN4 cells (gated on TCRβ$^+$ DN4 cells) (**h**) or γδTCR$^+$ DN cells (gated on DN thymocytes expressing γδTCR) (**I**); **h, i**, $n = 4$ mice each. See also Supplementary Fig. 4. For all graphs in (**a**–**c, e, f, h, i**), data are mean ± SD, **$p < 0.01$, ***$p < 0.001$ using two-sided Student's $t$ test. Source data are available for (**a**–**f, h, i**).

## Upregulation of PI3K/Akt signals in the absence of GFAT1 boosts growth and survival of αβ-T cells while increased ERK signaling skews lineage commitment towards γδ-T cells

To investigate alterations in signaling that could affect the viability of GFAT1-deficient cells, we analyzed basal phosphorylation of Akt, a key regulator of cell survival. The percentage of TCRβ$^+$-DN4 cells with Akt phosphorylated at Ser473, as well as the level of this phosphorylation (MFI) were significantly increased in the absence of GFAT1 as compared to WT cells (Fig. 5a). In contrast, there were no discernible changes in basal Akt-Thr308 phosphorylation in the same cell subsets with or without GFAT1 (Supplementary Fig. 5a). It is possible that increased Akt-Ser473 phosphorylation confers a biological advantage

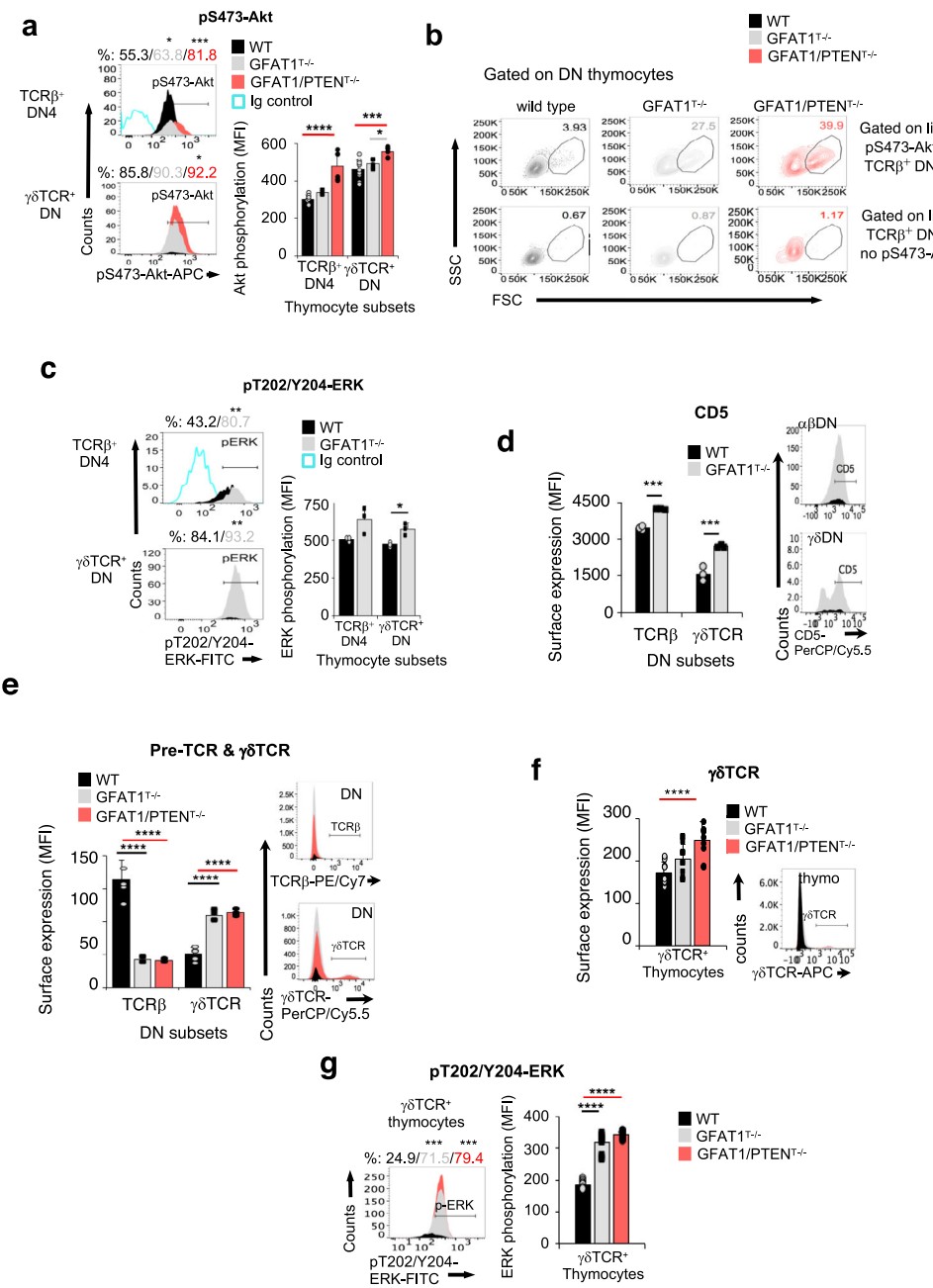

**Fig. 5 | Upregulation of PI3K/Akt signals in the absence of GFAT1 promotes growth and survival of αβ-T cells while increased ERK signals skew lineage commitment towards γδ-T cells.** Thymocytes from male and female WT, GFAT1- or combined PTEN/GFAT1$^{T-/-}$ mice were used for the following. **a–e** Thymocytes were stained for Lin, CD4, CD8α, CD25, CD44, CD5,TCRβ, and γδTCR, and intra-cellularly stained for pS473-Akt (**a, b**) or pT202/Y204-ERK (**c**). After flow cytometric analysis, proportion of cells that positively stained for each phosphorylated protein was plotted (**a–c**). Representative results out of three independent experiments with similar results are shown (**a–c**). Bar graphs (**a, c**) represent the amounts (median fluorescence intensity; MFI) of each specific phosphorylation ($n = 3$–9 mice each). Size (FSC) and granularity (SSC) of DN4 cells expressing TCRβ on their surface and with (pS473$^+$-Akt TCRβ$^+$-DN4) or without phosphorylated Akt at S473 (TCRβ$^+$-DN4 no pS473-Akt) was plotted (**b**). Numbers in each plot represent pro-portion of "blasting" cells. Bar graphs (**d, e**) represent the expression (median fluorescence intensity; MFI) of CD5 ($n = 3$ mice) (**d**) as well as either γδTCR or TCRβ (pre-TCR) ($n = 6$ mice)(**e**) on the surface of CD4$^-$CD8$^-$Lin$^-$ DN thymocytes. Repre-sentative FACS plots of the respective receptor staining out of three independent experiments with similar results are shown on the right side of the corresponding bar graph. **f, g** Thymocytes were surface-stained for γδTCR followed by phospho-ERK intracellular staining. A representative FACS plot of γδTCR staining (**f**) or phosphorylated-ERK (**g**) out of three experiments with similar results is shown. Bar graphs represent the median fluorescence intensity (MFI) of γδTCR surface expression (**f**) or phosphorylated ERK (**g**) ($n = 4$ mice each with 3 technical repli-cates). All bar graphs (**a–g**) denote mean ± SD. *$p < 0.05$, **$p < 0.01$, ***$p < 0.001$, ****$p < 0.0001$ using two-sided Student's $t$ test (**c, d**) or one-way ANOVA followed by Tukey's post-hoc test (**a, e–g**). See also Supplementary Fig 5. Source data are available for (**a–g**).

to a pool of GFAT1$^{T-/-}$ TCRβ$^+$-DN4 cells. Therefore, we gated on these cells and examined their size (FCS) and granularity (SSC). A higher proportion of pS473-Akt$^+$ TCRβ$^+$-DN4 cells lacking GFAT1 had a large size and increased granularity as compared to WT counterparts

(Fig. 5b). This distinct DN4 "blasting" subset was not found in the absence of Akt-Ser473 phosphorylation (Fig. 5a), suggesting that increased basal Akt-Ser473 phosphorylation could promote cell growth and survival in the absence of GFAT1. To further investigate this

notion, we analyzed how increasing PI3K/Akt signaling during GFAT1 deficiency could overcome the developmental defects in GFAT1$^{T-/-}$ mice. PTEN-deficiency resulted in markedly increased Akt-Thr308 and -Ser473 phosphorylation as well as augmented proportions of cells harboring this phosphorylated kinase, even in the absence of GFAT1 (Fig. 5a and Supplementary Fig. 5a, b). However, despite an even greater proportion of "blasting" cells among the pS473-Akt$^+$ TCRβ$^+$-DN4 subset (Fig. 5b), the constitutive Akt phosphorylation brought about by the loss of PTEN could not rescue αβ-T cell ontogeny in the absence of GFAT1. Indeed, combined GFAT1/PTEN-deficiency also chiefly delayed maturation to the αβ-lineage and skewed thymocyte development towards γδ-T cells (Supplementary Fig. 5c). Thus, these findings indicate that increasing PI3K/Akt signaling in the absence of GFAT1 is not sufficient to rescue αβ-T cell development. Additionally, since both *Gfat1* and *Pten* are excised by *Lck*-driven Cre recombinase, which is T cell lineage-specific, these results also validate that PTEN and GFAT1 are equally lost in both the αβ- and γδ-thymocytes. Indeed, Akt phosphorylation at both Thr308 and Ser473 were also significantly increased in GFAT1/PTEN$^{T-/-}$ γδTCR$^+$ DN thymocytes as compared to counterpart cells lacking only GFAT1 (Fig. 5a & Supplementary Fig. 5a). However, both genetically modified animal models had similar defects in αβ-T cell ontogeny that favored the development of γδ-T cells (Supplementary Fig. 5c). Thus, the observed skewing towards the latter lineage is not due to an incomplete *Gfat1* abrogation in those cells.

A strong TCR-triggered ERK activation is linked to γδ-lineage differentiation, while weak ERK signals favor αβ-thymocyte ontogeny[44,45]. In TCRβ$^+$-DN4 or γδTCR$^+$ DN cells lacking GFAT1, there was stronger basal ERK phosphorylation, higher proportions of phospho-ERK positive cells, and a corresponding increase in CD5 (Fig. 5c, d), which could thus favor differentiation towards the γδ-lineage. Consistent with a strong basal ERK signal, γδTCR was increased ~2x on the surface of GFAT1-deficient DN thymocytes of the γδ-lineage (Fig. 5e). This dramatic change in γδTCR expression in the absence of GFAT1 was mimicked by the combined absence of GFAT1 and PTEN (Fig. 5e). Notably, higher levels of γδTCR surface expression and basal ERK phosphorylation were observed for all thymocytes of the γδ-lineage lacking GFAT1 or GFAT1 and PTEN (Fig. 5f, g). These findings unravel that enhancing PI3K/Akt signals in the absence of GFAT1 boost growth and survival but not the development of αβ-T cells whereas increasing ERK signals skew lineage commitment towards γδ-T cells.

## Bypassing the GFAT1-catalyzed reaction by glucosamine supplementation rescues αβ-T cell development in fetal thymic organ cultures

If defective αβTCR levels account in part for the impaired ontogeny of αβ-T cells, we surmised that forced overexpression of this receptor may be sufficient in boosting this T cell subset while dampening the proportion of γδ-lineage. We therefore crossed either the WT or GFAT1$^{T-/-}$ mice with animals expressing the transgenic MHC-I restricted Vα2/Vβ5 TCR (OT-1 TCR). Enhancing the expression of αβTCR fully biased CD25$^-$ DN thymocyte differentiation towards the αβ-lineage, while abrogating γδ-T cell maturation even in the absence of GFAT1 (Supplementary Fig. 6a). However, OT-1 TCR overexpression did not restore normal DP nor CD8-SP maturation in OT-1/GFAT1$^{T-/-}$ transgenic mice (Supplementary Fig. 6b). Indeed, while DP thymocytes represented the majority of cells in WT OT-1 TCR transgenic mice, DN cells still largely dominated in GFAT1$^{T-/-}$/OT-1 thymi (Supplementary Fig. 6b). Similarly, mature CD8-SP thymocytes represent over 90% of CD8 expressing WT OT-1 subsets, but only ~10% of those cells when GFAT1 is abrogated in the transgenic OT-1 TCR mice (Supplementary Fig. 6c). Thus, forced surface overexpression of the αβTCR in the absence of GFAT1 is unable to restore normal αβ-T cell development. Nevertheless, increasing the subpopulation of thymocytes expressing

αβTCR on their surface restrained γδ-lineage ontogeny even when *Gfat1* was abrogated.

Since UDP-GlcNAc could be generated via salvage pathways, we probed whether nutrient supplementation could restore αβ-T cell development in FTOCs. We provided exogenous glucosamine (GlcN) in the culture media, a nutrient that could enter the HBP at a step distal to the GFAT1-catalyzed reaction (Supplementary Fig. 1b). DKG, an α-ketoglutarate analogue was also added as indicated, in order to boost the generation of intermediates needed for the production of UDP-GlcNAc as well as for the synthesis of nucleotides. Seven days of culture with the added metabolites restored the proportion of GFAT1-deficient thymocytes committed to the αβ-lineage close to WT levels (Fig. 6a). However, the percentage of GFAT1$^{T-/-}$ thymocytes expressing γδTCR did not decrease, but in fact also slightly increased, especially in the presence of GlcN & DKG added together. Further analysis revealed that GlcN alone or in combination with DKG augmented the expression of αβTCR on all αβ-lineage-committed thymocytes and in particular on SP cells, which normally are the subsets with high surface levels of this receptor (Fig. 6b, c). Metabolite supplementation also significantly increased the proportion of DP and SP thymocytes (Fig. 6d), while the amounts of CD8 coreceptor on either ISP, DP, or SP as well as CD4 on SP and DP were also restored to WT levels (Supplementary Fig. 6d, e). Furthermore, the relative number of the DN subsets lacking GFAT1 was restored closely to WT levels upon GlcN or GlcN/DKG supplementation (Fig. 6e), while the amounts of CD44 and CD25 expressed on the surface of GFAT1-deficient DN cells were also partially restored to WT levels (Supplementary Fig. 6f, g). Overall, our results show that bypassing the GFAT1-catalyzed reaction by the addition of glucosamine with or without DKG ex vivo can restore αβ-T cell development in GFAT1-deficient FTOCs.

## Dietary supplementation of GFAT1$^{T-/-}$ mice with glucosamine and α-ketoglutarate partially restores αβ-thymocyte development by increasing the viability of DN and SP cells

We next investigated if dietary supplementation with glucosamine and DKG could rescue αβ-T cell development in GFAT1$^{T-/-}$ mice. One month of GlcN/DKG-supplementation in vivo significantly increased the total number of GFAT1-deficient thymocytes although still at lower levels relative to WT animals (Fig. 7a). The proportions of αβ-lineage thymocytes were also restored to ~80% of the counterpart cells found in WT mice, with a corresponding decrease in the percentage of γδ-cells (Fig. 7b). There were ~5x more thymocytes expressing αβTCR in GlcN/DKG-incubated GFAT1-deficient cells (referred to as GD+) than without supplementation (Fig. 7c), while γδ-thymocyte numbers were modestly reduced in the GD+ mice (Supplementary Fig. 7a). Furthermore, the proportion of DN, CD8-ISP, DP and SP subsets were restored to almost WT levels in GD+ thymocytes (Fig. 7d). This was accompanied by a dramatic rise in the total number of DP cells in GD+ mice, albeit still lower than WT (Fig. 7e). CD8-ISP as well as CD4-SP and CD8-SP cell numbers were also significantly increased yet remained less compared to WT counterparts (Fig. 7e). The loss of GFAT1 dramatically decreased the proportion of live CD8-SP cells, which was remarkably well rescued in GD+ mice (Supplementary Fig. 7b). Viability of CD8-ISP and CD4-SP cells was also restored to WT levels upon metabolite supplementation. Thus, the augmented CD8-ISP, CD4-SP and especially CD8-SP cell numbers found in GD+ animals could be attributed to increased cell viability. αβTCR surface expression levels did not increase on DP cells of GD+ mice (Fig. 7f). Hence, metabolite supplementation unlikely improved their positive selection and differentiation to SP cells, which could partly explain the incomplete rescue of SP cell numbers in those animals. In contrast, αβTCR levels were restored to normal levels on CD4-SP and CD8-SP cells, which could contribute to their increased viability. Next, to determine how GlcN/DKG supplementation reprogrammed metabolism, we performed metabolomics analysis on the thymocytes of non-supplemented vs GD+ mice. A slight but significant

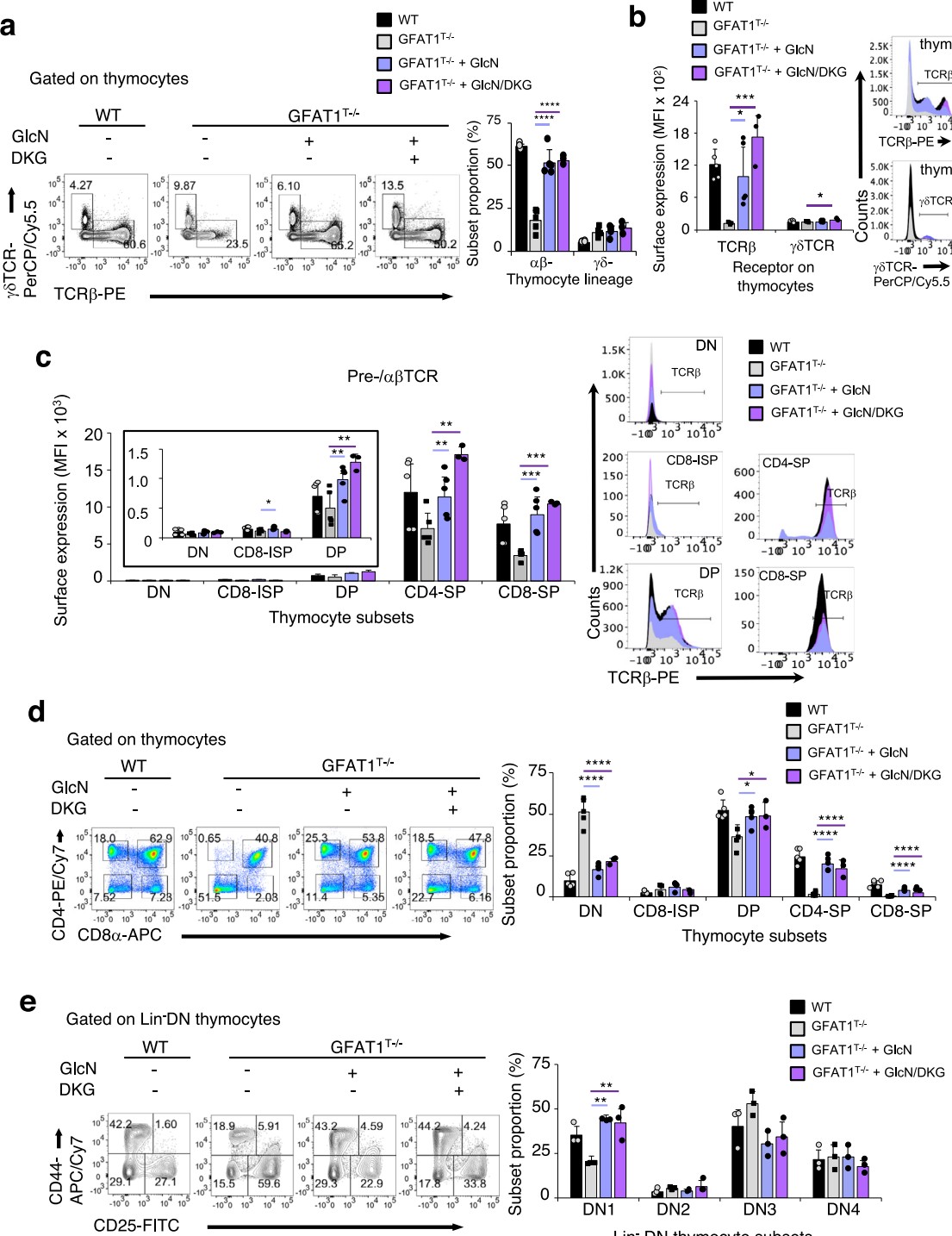

**Fig. 6 | Glucosamine supplementation rescues αβ-T cell development in fetal thymic organ cultures.** WT or GFAT1[T−/−] mice were used for the following experiments. Thymic lobe from fetus was incubated in complete media containing glucosamine (GlcN) with or without dimethyl-2-ketoglutarate (DKG) and the adjoining lobe was cultured in complete media only as control. Thymocytes were stained for CD4, CD8, CD25, CD44, TCRβ, and γδTCR and analyzed by flow cytometry. **a** Plot and bar graph represent proportion of subsets expressing either surface TCRβ (αβ-lineage) or γδTCR (γδ-lineage) after 7 days of culture ($n = 3–5$ fetal lobes). Plot is representative of three experiments with similar results (**a**). **b** Bar graph showing TCRβ and γδTCR levels (median fluorescence intensity; MFI) on the surface of αβ- and γδ-lineage-committed thymocytes, respectively ($n = 3–5$ fetal lobes) and representative FACS plots of the respective receptor staining from three experiments with similar results are shown. **c** Bar graphs (median fluorescence

intensity; MFI) ($n = 3–5$ fetal lobes) and FACS plots of TCRβ levels expressed on the surface of each subset (**c**) The inset is a blowup of TCRβ expression on DN to DP stages. FACS plots are representative of three experiments with similar results. **d** Thymocyte subsets expressing either CD4 or CD8 were plotted. FACS plots are representative of three experiments with similar results. Bar graphs show proportion of different thymocyte subsets ($n = 3–5$ fetal lobes). **e** Immature CD4−CD8−Lin− DN thymocytes were stained for CD25 and CD44 expression. Representative FACS plots out of three experiments with similar results showing relative number of DN subsets are indicated in each quadrant and plotted in the bar graph ($n = 3$ fetal lobes). All graphs from (**a–e**) are mean ± SD. *$p < 0.05$, **$p < 0.01$, ***$p < 0.001$, ****$p < 0.0001$ using one-way ANOVA followed by Tukey's (**a**, **b**) or Šidák's (**c–e**) post-hoc test. See also Supplementary Fig. 6. Source data are available for (**a–e**).

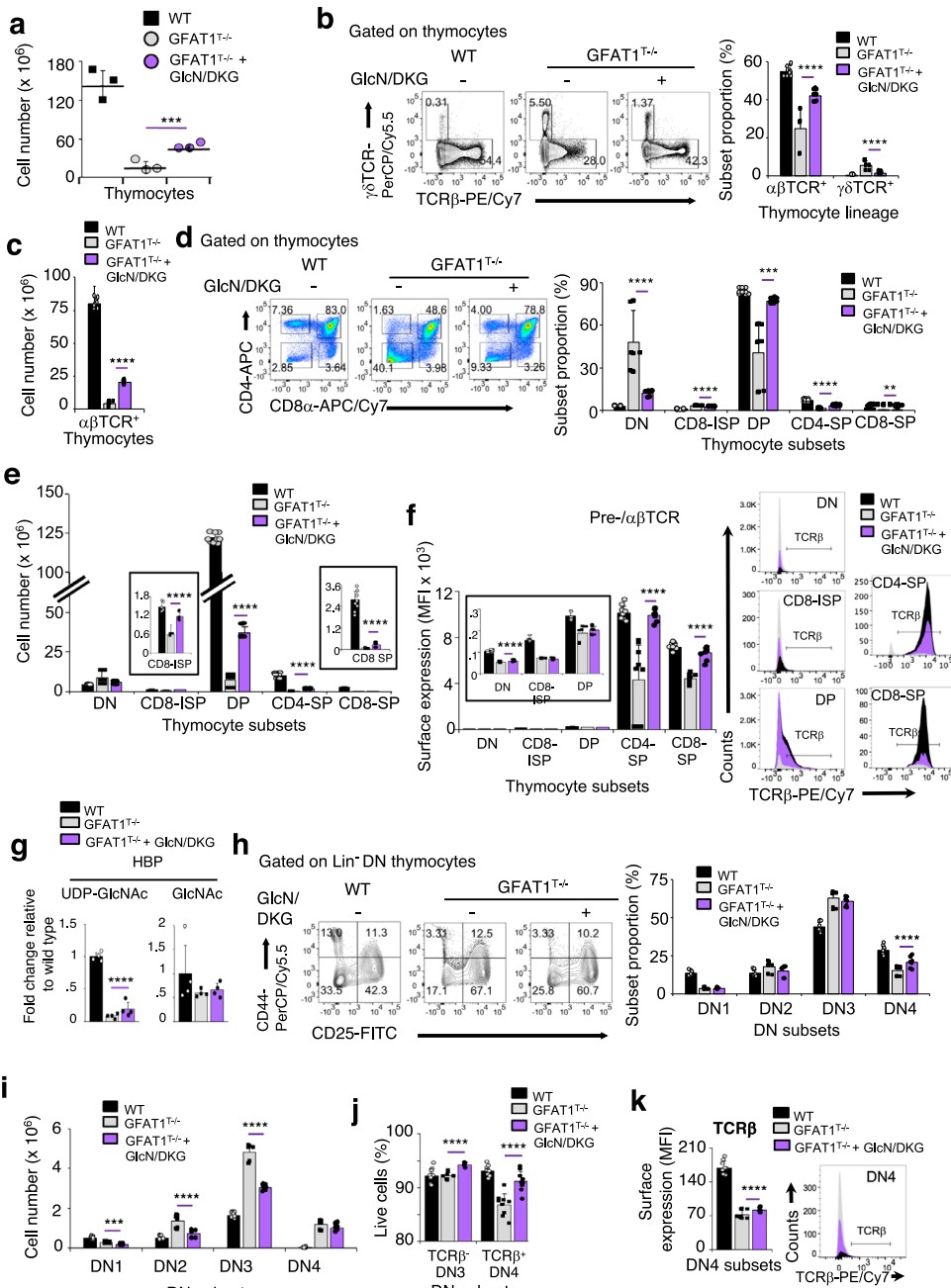

**Fig. 7 | Dietary supplementation of GFAT1[T−/−] mice with glucosamine and α-ketoglutarate partially restores αβ-thymocyte development by increasing the viability of DN and SP cells.** Three-week-old male and female GFAT1[T+/+] (WT) or GFAT1[T−/−] littermates were fed with regular water or water containing GlcN and DKG (100 mM each) (GlcN/DKG-supplemented GFAT1[T−/−] mice are also referred to as GD+) and thymocytes were harvested after 1 month. Cells were stained for Lin, CD4, CD8, CD25, CD44, Annexin V, TCRβ and γδTCR and analyzed by flow cytometry (**a–f**, **h–k**) or subjected to metabolite analysis (**g**). **a** Total cell number for each mice are indicated (*n* = 3 mice each). **b**, **c** Proportion of subsets expressing either high TCRβ or high γδTCR on their surface are shown in representative plots from three experiments with similar results and bar graph (**b**). Absolute numbers of αβ-lineage cells are shown (**c**). (**b**, **c**, *n* = 3 mice with 3 technical replicates each). See also Supplementary Fig. 7a. **d**, **e** Thymocyte subsets expressing either CD4 or CD8 were plotted. FACS plots are representative of three independent experiments with similar results. Bar graphs represent the proportion (**d**) and absolute number (**e**) of each subset (**d**, **e**, *n* = 3 mice with 3 technical replicates each). Insets are blowups of the absolute number of CD8-ISP and CD8-SP cells. **f** Bar graph (median

fluorescence intensity; MFI) of TCRβ expressed on the surface of each thymocyte subset with corresponding FACS plots. Inset is a blowup of TCRβ expression on DN to DP stages (*n* = 3 mice each with 3 technical replicates each). Representative FACS plots are from three experiments with similar results. **g** Metabolites were isolated from thymocytes and analyzed by LC/MS. Bar graphs represent fold changes of indicated metabolite relative to WT (*n* = 4 independent samples each). See also Supplementary Fig. 7c. **h**, **k** Immature CD4−CD8−Lin− DN thymocytes were stained for CD25 and CD44 expression. Relative subset numbers are indicated in each quadrant and plotted in the bar graph (**h**). Representative FACS plots are shown from three experiments with similar results. Total cell numbers from each DN subset were plotted (**i**). The viability of TCRβ−DN3 and TCRβ+-DN4 was plotted (**j**). Bar graph of TCRβ expression (median fluorescence intensity; MFI) on the surface of DN4 thymocyte and representative FACS plot from three experiments with similar results are shown (**k**). *n* = 3 mice each with 3 technical replicates each (**h–k**). All graphs from (**a–k**) denote mean ± SD. **\**p* < 0.01, **\*\**p* < 0.001, **\*\*\**p* < 0.0001 using one-way ANOVA followed by Šidák's (**a–f**, **j**, **k**) or Tukey's (**h**, **i**) post-hoc test. Source data are available for (**a–k**).

increase in UDP-GlcNAc but not GlcNAc was observed in GD+ mice (Fig. 7g). Interestingly, purines and pyrimidines trended towards WT levels in GD+ mice (Supplementary Fig. 7c). Hence, in the absence of *dn*-HBP, GlcN/DKG in the diet can partially restore UDP-GlcNAc and nucleotide levels. Together, our findings indicate that supplementation with GlcN and DKG in the absence of GFAT1 rescues the proportion of thymocyte subsets but does not fully restore DP cell number, nor improve positive selection. Instead, dietary supplementation augments αβTCR surface expression and consequently survival of SP thymocytes.

Next, we analyzed whether GlcN/DKG supplementation had an effect on the DN subsets. In contrast to metabolite supplemented GFAT1$^{T-/-}$ FTOC (Fig. 6e), the relative number of DN3 cells remained unchanged in GD+ as compared to GFAT1$^{T-/-}$ mice, whereas the proportion of DN4 cells modestly but significantly increased (Fig. 7h). When cell numbers were calculated, we found that all DN subsets in GD+ mice were closer to the numbers found in WT animals (Fig. 7i), suggesting an effect of the metabolites on early thymocyte ontogeny. Similar to CD8-ISP and CD4- or CD8-SP cells (Supplementary Fig. 7b), we found that GlcN/DKG supplementation boosted the viability of DN3 cells that yet lack TCRβ expression as well as induced a more dramatic survival of TCRβ$^+$DN4 cells as compared to their non-supplemented GFAT1-deficient counterparts (Fig. 7j). However, TCRβ expression was only marginally rescued on the surface of GD+ DN4 subsets (Fig. 7k), suggesting that this receptor probably has minimal impact on their increased survival. Overall, our experiments reveal that dietary GlcN/DKG supplementation in mice improves, but not fully rescues, thymocyte development that was compromised by the loss of GFAT1.

## Discussion

Our studies reveal the requirement for GFAT1-mediated de novo hexosamine biosynthesis to meet the increased demand for this metabolite during the highly proliferative stages of αβ-T cell ontogeny. Although γδ-T cells undergo relatively slower proliferation than αβ-T cells and thus have less dependence on GFAT1 during development, we found that they rely on GFAT1 to generate γδ-TCR diversity. The absence of *dn*-HBP decreased UDP-GlcNAc and was accompanied by defective *N*-glycosylation, enhanced TCR misprocessing and UPR. Supplementation with the salvage metabolite glucosamine mostly rescued the development of FTOCs ex vivo. It also restored αβ-T cell numbers in vivo due to enhanced viability of DN and SP cells but poorly rescued the developmental block during the highly proliferative DN3-DN4 and CD8-ISP stages. Thus, our findings unravel the requirement for sufficient hexosamines for cell survival and to allow robust expansion and maturation of αβ-thymocytes. Our findings also underscore the distinct metabolic programs supporting αβ- vs γδ-T cell development.

The requirement for GFAT1 during αβ-T cell development is particularly critical during the DN3b-DN4 and CD8-ISP stages. In highly proliferating cultured cells, maintaining or increasing GFAT1 expression is necessary during glutamine- or glucose-limiting conditions[37,38]. Here, we unravel that GFAT1 or the *dn*-HBP is specifically required at the highly proliferative stages of αβ-T cell ontogeny and for the survival of SP thymocytes. The robust expansion of DN3b/DN4 and CD8-ISP thymocytes would increase the demand for nutrients and metabolites. Sufficient UDP-GlcNAc would be necessary to meet the needs of highly proliferating αβ-lineage thymocytes in order to properly generate a diverse repertoire of pre-TCR/αβTCR[46,47]. The TCRβ and TCRα chains experience diversification during the DN3b-DN4 and CD8-ISP stages, respectively, followed by clonal expansion of those cells[48]. The increased receptor synthesis during these proliferative phases would thus escalate the demand for UDP-GlcNAc. Limiting UDP-GlcNAc levels would dampen proper glycosylation, folding and assembly of receptor complexes such as the pre-TCR or αβTCR. Indeed, we found augmented intracellular misprocessing of TCRβ in

the absence of GFAT1. Moreover, *N*-glycomics profiling revealed increased proportion of oligomannose/paucimannose glycans whereas the percentage of complex/hybrid-type was decreased. Defective levels of these glycan moieties have been found in diseases associated with protein misfolding[49]. Results from our biochemical studies also point to increased ER stress as indicated by enhanced phosphorylation of eIF2α and increased IRE1α expression. In support of our biochemical data, proteomics analysis revealed the upregulation of proteins involved in the UPR as well as tRNA charging, which is linked to the ISR[41]. Together, these findings are consistent with more rampant defects in *N*-glycosylation due to limited UDP-GlcNAc in the absence of GFAT1, leading to ER stress[50,51]. Our findings bolster previous studies on the importance of proper *N*-glycosylation to maintain proteostasis and normal T cell development and function[52,53]. However, it is important to note that certain receptors, such as the cytokine receptors CD25 and CD127 were upregulated at the surface of GFAT1-deficient DN cells despite limiting amounts of UDP-GlcNAc. The successful glycosylation and cell surface routing of such glycoproteins would be biased towards the more abundant nascent polypeptides[46]. Consistent with this notion, we found a dramatic elevation of CD25 mRNA, whereas TCRβ mRNA levels were diminished in the same GFAT1-deficient thymocytes. Although defective αβTCR surface expression could contribute significantly to impaired thymocyte development, its overexpression was unable to rescue proper αβ-T cell ontogeny in the absence of GFAT1. This is likely due to impaired expression of other cell surface proteins, such as Notch1 or CXCR4, that also coordinate thymocyte development. UDP-GlcNAc is also critical for *O*-GlcNAcylation of cytosolic proteins, which was significantly decreased in αβ-thymocytes lacking GFAT1. In this regard, it is noteworthy that deficiency of enzymes involved in regulating protein *O*-GlcNAcylation, also leads to aberrant early thymocyte development[15,18]. Future studies should address how the absence of *dn*-HBP could affect specific *N*- and *O*-GlcNAcylation of glycoproteins that impact T-cell development.

Why do γδ-T cells have increased expansion despite aberrant γδ-TCR diversity in the absence of GFAT1? One possibility is that since they have intrinsic slower proliferation[30,54], the relatively less demand for hexosamines can be fulfilled by salvage mechanisms. This difference in metabolic needs compared to developing αβ-T cells is further supported by the finding that γδ-T cells, unlike αβ-T cells, are independent of mTOR signaling[10,55], which controls several de novo biosynthetic pathways including the HBP[37,56]. Despite increased γδ-T cell numbers in the absence of GFAT1, γδ-TCR diversity was compromised. It is plausible that producing a diverse TCR repertoire requires abundant hexosamines to facilitate proper folding such that even γδ-T cells poorly generate new TCR chains and instead clonally expand early immature isotypes. Differences in TCR chain glycosylation could also account for distinct need for UDP-GlcNAc amounts. Such a difference in glycan complexity has been recently documented for the TCR expressed on Th17 cells[16]. It is also possible that altered *N*-glycosylation of the TCR could impair their signaling, thus affecting T cell activation and differentiation[53,57–59]. How γδ-TCR diversity is dependent on *dn*-HBP would need further investigation. γδ-T cell development is also favored in the presence of high IL7R signaling[60]. The increased levels of CD127 (IL7Rα) on the surface of GFAT1-deficient DN cells as compared to WT counterparts could therefore contribute to skewing thymocyte development towards the γδ-lineage. The precise metabolic processes that drive γδ-T cell development and generation of γδ-TCR as well as αβ-TCR diversity remain to be further investigated.

We also demonstrate that GFAT1-deficiency shares similar developmental defects with mTORC2 disruption including partial blocks at the DN3-DN4 and CD8-ISP stages as well as reduced αβTCR expression[27]. The metabolic and proteomic aberrations during rictor deletion are somewhat less profound compared to GFAT1 abrogation. Rictor ablation was also reported not to affect γδ-T cell cellularity

whereas raptor deletion increased γδ-T cell proportion relative to αβ-T cells[55], similar to GFAT1 deficiency. However, unlike GFAT1 ablation, mTORC1 disruption does not alter the composition of various Vγ chains[55]. Thus, although mTORC1 and mTORC2 are involved in various aspects of metabolism and are crucial for early T cell development[27,40,55,61–63], loss of either mTORC may not be sufficient to profoundly affect metabolism and development. As the mTORCs respond to nutrients[56], the limitation of a specific nutrient such as hexosamines would likely feedback to the mTORCs to modulate its signaling. Consistent with this notion, we found that the expression of rictor was significantly reduced in GFAT1$^{T-/-}$, suggesting that UDP-GlcNAc could modulate mTORC2. In turn, mTORC2 responds to the limitation of glucose or glutamine to modulate GFAT1[37,38]. The mechanisms underlying this feedback regulation between the HBP and mTORCs deserve further interrogation.

The hexosamine salvage metabolite, glucosamine, can appreciably rescue the developmental defects of thymocytes during ex vivo culture of fetal thymi. Interestingly, it can only partially rescue those defects when added together with DKG as a dietary supplement to 3-week-old mice, suggesting that the age of the animal plays a role in this process. It is noteworthy that the percentage of DN thymocytes committed to the γδ-lineage dominate over those constrained to the αβ-lineage in WT newborns, while the reverse proportions are found in 1-week-old WT mice and thereafter. The distinct diets that embryos and post-birth mice encounter may possibly dictate in part lineage commitment of developing thymocytes. Interestingly, all the different thymocyte subsets that were altered by the absence of GFAT1, as well as the expression of critical receptors on their surface were restored to WT levels or beyond following ex vivo supplementation of FTOC with GlcN or GlcN & DKG. In contrast, 1 month diet supplementation of 3-week-old GFAT1$^{T-/-}$ mice only modestly improved αβ-T cell ontogeny by mainly increasing the viability of certain subsets such as DN4, CD8-ISP as well as both, CD4- and CD8-SP cells. The expression of αβTCR on the surface of the latter thymocytes was also rescued to the levels found on WT cells, suggesting that increased receptor amounts could account in part for the augmented cell viability. Consistent with the incomplete rescue of αβ-T cell developmental defects upon dietary supplementation, metabolomics analysis revealed a slight but statistically significant increase in UDP-GlcNAc in GlcN/DKG-supplemented GFAT1$^{T-/-}$ mice. These findings imply that salvage mechanisms and/or the GlcN administered dose may be inefficient in generating ample UDP-GlcNAc to meet the high demand of developing αβ-lineage thymocytes. Interestingly, the levels of nucleotides and nitrogenous bases were significantly restored in these mice and could thus contribute to increased survival during development. Hence, sufficient hexosamine levels, achieved via de novo biosynthesis, are indispensable for proper early T cell development. Whether GFAT1 could play other crucial roles, independent of its catalytic activity during development also remains a possibility. Collectively, our findings unravel how dietary manipulations that boost the levels of critical metabolites such as hexosamines can be beneficial in reprogramming lineage commitment during early T cell ontogeny.

## Methods

### Ethics statement

Handling and experimentation protocols for all mouse experiments have been reviewed, approved, and used in accordance with the guidelines of the Institutional Animal Care and Use Committee of Rutgers University (PROTO999900133 and PROTO999900132).

### Mice and immunoblotting

Mice were housed in an environmentally controlled, pathogen-free barrier facility on a 12-h light-dark cycle, ambient temperature was 20–26 °C and humidity was maintained between 40–60%. Standard rodent chow diet (PicoLab Enerpro, Cat#1816346, TestDiet, MO) and water were available *ad libitum*. Experimental and control animals were co-housed and bred separately. Mice were euthanized by carbon dioxide asphyxiation followed by cervical dislocation. Homozygous C57BL/6 *Gfat1*$^{f/f}$ (Mary Lyon Center Harwell Science and Innovation Center, UK;Stock id: EM:07321;Code: GFPT1-PL-TM1C-FLPE-SEG and B6NTAC-USA), C57BL/6 *rictor*$^{f/f64}$ or C57BL/6 Pten$^{f/f}$ (Jackson Labs; Strain# 006440;129S4-Ptentm1Hwu/J) mice were crossed with heterozygous C57BL/6 Lck-Cre$^{+/-}$ animals [Taconic farms, NY; Model# 4197;B6.Cg-Tg(Lck-Cre)1CwiN9], which generates WT (*Lck-Cre*$^{-/-}$/*Gfat1*$^{f/f}$, Lck-Cre$^{-/-}$/*rictor*$^{f/f}$, Lck-Cre$^{-/-}$/*Pten*$^{f/f}$ as well as T-cell-specific *Gfat1* (*Lck-Cre*$^{+/-}$/*Gfat1*$^{f/f}$, aka GFAT1$^{T-/-}$), *rictor* (*Lck-Cre*$^{+/-}$/*rictor*$^{f/f}$, aka rictor$^{T-/-}$) or *Pten* (*Lck-Cre*$^{+/-}$/*Pten*$^{f/f}$, aka PTEN$^{T-/-}$) knockout mice owing to the expression of Cre under the control of the proximal promoter of Lck. GFAT1/PTEN$^{T-/-}$ double knockout mice were obtained by cross-breeding GFAT1$^{T-/-}$ and PTEN$^{T-/-}$ animals for several generations. To specifically delete GFAT1 in the OT-1 TCR background, we crossed C57BL/6/OT-1 mice[65] with homozygous GFAT1$^{T-/-}$ to obtain OT-1/GFAT1$^{T-/-}$ animals. All mice were genotyped by PCR using the respective primers (synthesized commercially by Integrated DNA Technologies, IA) described in Supplementary Table 1. To perform biochemical experiments (immunoblotting, metabolomics, proteomics and glycomics), more samples were acquired from GFAT1-deficient mice due to lower thymocyte numbers relative to WT. Typically, the amount of thymocytes harvested from a 5-week-old WT thymus is equivalent to about 2–4 thymus from rictor-deficient and 3–5 thymus from GFAT1-deficient (or any combination with GFAT1-deficient) thymus. Tissues were removed from littermates, micro-sliced and resuspended in RIPA buffer (50 mM Tris-HCl pH8.0, 100 mM NaCl, 5 mM EDTA, 1% Triton X-100, 0.2% SDS, 0.5% Na-deoxycholate supplemented with protease inhibitors). Protein concentrations were determined by Bradford assay. About 30 μg of extracts from each sample were resolved by SDS-PAGE, transferred onto PVDF membranes followed by immunoblot analysis using antibodies described in Supplementary Table 2. Antibodies were used at 1:1000 dilution except for β-actin (1:5000). Antibodies were validated by the corresponding vendor. ImageJ software [NIH; version 1.51 (100)] was used for densitometric analysis of protein expression or phosphorylation. Throughout the manuscript, MFI refers to Median Fluorescence Intensity.

### Thymocyte immunostaining, gating strategy, proliferation and viability

Thymocytes and peripheral T cells harvested in complete DMEM from thymus and spleen, respectively were counted by trypan blue exclusion. Cells were stained for receptor expression using antibodies listed in Supplementary Table 3. Antibodies were validated by the corresponding vendor and used at 1:100 to 1:400 dilution for cell surface and 1:50 or 1:100 for internal staining. Stained cells were analyzed using a BD FACS Verse flow cytometer and FlowJo (TreeStar, OR) software (Version V10.0.8) or further fixed and permeabilized with BD Cytofix/Cytoperm reagents according to the manufacturer's protocol. Permeabilized cells were then stained for intracellular protein expression, phosphorylation or O-GlcNAcylation and analyzed by flow cytometry. For Lin⁻ DN thymocyte gating, we excluded cells expressing the lineage markers NK1.1 (NK cell marker), B220 (B cell marker), Ter119 (erythroid marker), Gr1 (neutrophil marker), CD3ε (T cell Marker). DN1 = CD25⁻CD44⁺, DN2 = CD25⁺CD44⁺, DN3 = CD25⁺CD44⁻, DN4 = CD25⁻CD44⁻. The total (absolute) number of cells was calculated by multiplying the cell counts by the relative number of subsets in a pool of cell. To analyze cell proliferation, thymocytes were labeled with 2 μM of CFSE (Sigma-Aldrich) prior to culturing in complete DMEM media at 37 °C for up to 48 h. Cells were collected at 0, 24 or 48 h, stained and cell division was analyzed on the flow cytometer by monitoring the shift in CFSE fluorescence. To assess cell viability, ex vivo-cultured thymocytes were stained with Annexin V (BD

Pharmingen, CA) prior to flow cytometric analysis. See Supplementary Figs. 8–10 for detailed gating strategy.

## Lectin pull-down

For lectin binding assays, thymocytes from OT-1/WT and OT-1/GFAT1[T–/–] mice were harvested, counted, and cultured at 37 °C in complete DMEM with or without 50 μM of the proteasome inhibitor, MG132. After 4 h of culture, thymocytes were lysed using RIPA buffer. 300 μg of lysates were incubated overnight at 4 °C with 20 μL of lectin-agarose (Vector laboratories, CA), followed by washing with PBS buffer containing 0.25% TX-100. Whole cell lysates or pull-down aggregates were fractionated on SDS-PAGE followed by immunoblot analysis of TCRβ expression.

## Metabolite Analysis and Quantitative Proteomics

Equivalent thymocyte numbers (~100 million cells each) from 5-week old WT, GFAT1[T–/–] and rictor[T–/–] mice or 7-week old animals that were fed regular drinking water or water that was supplemented with 100 mM glucosamine (GlcN) & 100 mM dimethyl-α-ketoglutarate (DKG) for a month were harvested and intracellular metabolites were extracted by cold methanol extraction (40:40:20 methanol:acetonitrite:water with 0.5% formic acid. After 5 min incubation, 15% $NH_4HCO_3$ was added and cell lysate/methanol mixture was centrifuged at 15,000 g for 10 min at 4 °C. Supernatants were recovered and samples were analyzed at the Rutgers Metabolomics Shared Resource using HILIC-LC-MS. HILIC separation was performed on a Vanquish Horizon UHPLC system (Thermo Fisher Scientific, Waltham, MA) with an XBridge BEH Amide column (150 mm × 2.1 mm, 2.5 μm particle size, Waters, Milford, MA). MS scans were obtained on Q Exactive PLUS Orbitrap mass spectrometer (Thermo Scientific) in negative ionization mode with a resolution of 70,000 at m/z 200, in addition to an automatic gain control target of $3 × 10^6$ and m/z scan range of 72 to 1000. Metabolite quantitation data were obtained using the Maven software package (Build 682). Thymocytes from 3 WT, 6 GFAT1[T–/–] and 6 rictor[T–/–] mice were harvested, pooled into duplicates, lysed in RIPA buffer, and subjected to quantitative proteomics (Rutgers Biological Mass Spec Facility). Cell lysates were digested with trypsin and labeled with TMT10 reagent. Pooled peptides were fractionated by alkaline reverse phase chromatography and fractions were analyzed by nanospray LC-MS/MS using a Thermo Q Exactive HF mass spectrometer using previously described instrument settings, peak list generation and database searching[66]. Extracted TMT reporter ion intensities for each protein were compared from mutants and WT to measure ratio changes, and the significance of changes was determined using $t$ test (Microsoft Excel, version16.16.23). Proteomics data were analyzed using Ingenuity Pathway Analysis software (Qiagen; Version 01-20-04).

## N-glycomics

Thymocytes from 5–6-week-old OT-1/WT (pooled from 3 mice) or OT-1/GFAT1[T–/–] mice (pooled from 6 mice) were harvested and submitted to the CCRC (Athens, GA) for analysis. Cells were mixed with 50 mM ammonium bicarbonate buffer, passed through 26-gauge needle with syringe and probe sonicated. The proteins were reduced using denaturation buffer (NEB) and desalted using 10 kDa centrifuge filters and probe sonicated. The N-glycans were released from the samples by treating with PNGaseF (37 °C, 24 h). Released N-glycans were permethylated using iodomethane and DMSO/NaOH base. Dried samples were dissolved in DMSO and methylated using DMSO/NaOH base and iodomethane. The reaction was quenched with water and glycans extracted with dichloromethane and dried. Dried glycans were re-dissolved in MeOH and profiled using AB Sciex MALDI-TOF/TOF MS 5800 system in reflector positive mode. The N-glycan structures were assigned using GlycoWorkbench software (version 2.0) based on precursor masses (Sodiated) and common mammalian biosynthetic pathways.

## Quantitative real-time PCR

Thymocytes were harvested from 5-week-old WT or GFAT1[T–/–] littermates. RNA was extracted from $1×10^7$ thymocytes/point using the RNeasy kit (Qiagen, CA). Reverse transcription reactions were performed using the High-Capacity cDNA Reverse Transcription Kit (4368814, Applied Biosystems). The resulting cDNA products were amplified by quantitative real-time PCR using the respective primers described in Supplementary Table 1 and PowerUp SYBR Green Master Mix (A25778, Applied Biosystems) on a QuantStudio 3 Real-Time PCR System (Applied Biosystems). Relative expression levels were normalized using *Gapdh* or *tubulin* as a reference control.

## Metabolite supplementation

E18-19 embryo were removed from pregnant mice and tail snips were collected for genotyping, while their thymus was dissected under microscope for fetal thymic organ culture. One of the two thymic lobes was cultured in complete DMEM supplemented with 5 mM GlcN alone or together with 10 mM DKG, while the other lobe was cultured, as control, in complete DMEM. Media with or without supplement was replenished every other day. After 7 days of culture, thymocytes were extracted from FTOCs, stained, and analyzed by flow cytometer. For in vivo diet supplementation, 3-week-old WT or GFAT1-deficient littermates were fed *ad libitum* with regular water or water containing GlcN and DKG (100 mM each). Water with or without metabolite supplementation was changed every week for 1 month. Thymocytes were harvested, stained, and analyzed by flow cytometer. For metabolite analysis, equivalent thymocyte numbers were pooled for each condition and analyses was conducted on quadruplicates.

## Statistics and reproducibility

All statistical analyses were performed using Microsoft Excel version 16.16.23 for Student's $t$ test and GraphPad Prism (version 9.4.1 [458]) for ANOVA and *post-hoc* test. No statistical methods were used to predetermine the sample size. Each experiment was conducted with biological replicates (at least two mice per group) and repeated multiple times (at least two independent experiments). Replicate experiments were reproducible and no data were excluded from the analyses. The experiments were not randomized. The investigators were not blinded to allocation during experiments and outcome assessment except for the FTOC supplementation experiments wherein genotype was determined only at the end of the experiment. All data from graphs represent mean and error bars denote SD and statistical relevance was determined by two-sided Student $t$ test or one-way ANOVA with Tukey's or Sidaks's *post-hoc* test as appropriate.

## Reporting summary

Further information on research design is available in the Nature Portfolio Reporting Summary linked to this article.

## Data availability

All data are included in the Supplementary Information or Source Data file. The raw numbers for charts and graphs are available in the Source Data file whenever possible. The proteomics datasets have been deposited in the MassIVE database under the accession code MSV000089842 Source data are provided with this paper.

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

## Acknowledgements
The authors thank Drs. Michael N. Hall and Markus Ruegg for sharing the rictor^fl/fl mice, Drs. Peter Lobel, David Sleat and Haiyan Zheng for help with the quantitative proteomics, Drs. Parastoo Azadi and Asif Shajahan for glycomics analysis, Dr. Audrey Minden and Nikhil Patel for assistance with IPA analysis, Janet Wei and Maya Aleksandrova for assistance with mice genotyping, Jay Kavia, Christian Kim and Tatiana Hernandez for figure preparation. Glycomics analysis was performed at the Complex Carbohydrate Research Center (CCRC) and was supported in part by the National Institutes of Health (NIH)-funded R24 grant (R24GM137782) to Dr. Azadi. This work was supported by NJCCR grant COCR22PRG009 (E.J.) and fellowship DCHS20PPC010 (L.T.), COCR22PDF002 (V.dS.), LLS Scholar Award 1386-23 (D.H.), NIH Grants P30CA072720-5923 (X.S.), R01CA236936 (D.H.), and R01GM137493 (E.J.).

## Author contributions
G.W. and E.J. conceived and designed the study. G.W., M-L.L., L.T., V.dS., X.S., D.H., and E.J. designed and performed the experiments as well as analyzed the results. M-L.L. performed lectin pull-downs, L.T. performed qRT-PCR analysis, V.dS. conducted studies on thymocyte numbers under the direction of D.H. X.S. conducted metabolomics by LC-MS. E.J. performed IPA on proteomics results. All other experiments were performed by G.W. E.J., and G.W. wrote the manuscript, which was edited by all authors.

## Competing interests
The authors declare no competing interests.
