## [Peer Review File · Nature Communications]

REVIEWER COMMENTS

Reviewer #1 (Remarks to the Author):

This study examines the molecular mechanisms that control the development of abT-cells and gd-cells in the thymus. Given the multiple roles that these cells play in regulation and function of the immune response, studies of this kind are important, as understanding of intrathymic T-cell development remains incomplete. Specifically, the authors have examined the role of metabolism and the hexosamine biosynthesis pathway (HBP) in this process. While the importance of this pathway and the events that it controls are relatively well studied in other biological systems, there is little evidence for its importance in the context of thymus biology. As such, the study is of interest and timely.

The authors show that use of mice carrying a T-cell specific deletion of Gfat1 (the rate limiting enzyme for de novo HBP synthesis) impairs ab but not gdT-cell development. This maps to alterations in the expression of multiple cell surface receptors known to play a role in T-cell development, including TCR, Notch and CXCR4. Finally, the authors show that dietary supplementation of T-cell deficient Gfat1 mice, and in vitro supplementation of FTOC, can overcome defective T-cell development. Overall, the study is of interest, and contains novel findings that open up further study of new pathways relevant to intrathymic T-cell development. There are several issues that need to be addressed:

1. In their study of T-cell specific Gfat1 deficient mice, the authors use p56lckCre to drive deletion in thymocytes. While this is a fine model, the authors must use Cre only mice as controls, rather than the WT mice that are currently used (eg Fig1). This is because P56lckCre alone has previously been shown to result in thymus defects (Shi and Petrie Plos One 2012) which may well influence the degree of difference between control and experimental mice currently being used,
2. Can the authors include examples of FACS plots for CD27, Notch 1, CD98, GluT1, O-GlcNAcylation that are shown as MFI (eg Figures 2, 3)?
3. Regarding comparative analysis of abT-cell development with gdT-cell development, it is important to note that in rescue experiments in vivo (fig 6) or in vitro (fig 5), the nature of the gdT-cells, and their developmental requirements, will be distinct. For example, in in vitro FTOC experiments: gdT-cells will be synchronised waves of cells expressing fetal invariant gdTCRs (eg dendritic epidermal T-cells), while in vivo experiments will analyse only adult gdT-cell development where this tissue restriction/limited TCR diversity is not evident. Do the authors have data on how fetal and adult gdT-cells are influenced by absence of Gfat1? Analysis of fetal/neonatal P56LckCreGfat1 floxed mice may go part way to addressing this.

Reviewer #2 (Remarks to the Author):

This is a very interesting and novel study that shows the importance of GFAT1 and hexosamine biosynthesis in T-cell development. This is timely and of general interest to the community.

However, I believe the manuscript requires major revision to be ready for publication, particularly in relation to the analysis of thymocyte subsets and T-cell development

-throughout the figures, the term "relative cell number (%)" is used to label bar chart y-axis which in fact show percentages (not relative cell number). This is confusing and the focus on comparison of the percentage or proportion of a thymocyte population rather than the number of cells in the population is confusing and misleading. The analysis of number of cells clearly shows that the number of DN cells is normal, but a partial developmental arrest occurs from ISP onwards (Fig. 1b, h, j). Therefore, the fact that the percentage of DN cells is increased is not very relevant, and is confusing.

-analysis of DN subsets: the manuscript states that the DN subsets are gated as Lin-neg and staining includes CD4, CD8, CD3e, NK1.1, B220, Ter119, Gr1. Presumably this means that CD3e+

cells have been gated out of the analyses of the data shown in Fig2 (excluding NKT-cells, gammadelta cells); however, if CD3 has been gated out, then cell surface TCRbeta+ cells should also have been excluded, but Fig. 2e, h, show around 30% WT DN4 cells express cell surface TCRbeta. This does not seem right: at this stage of abT-cell development expression of the pre-TCR complex (TCRbeta with the chains of CD3) is very low but icTCRbeta expression is easy to detect. Additionally the surface TCRbeta staining on WT between Fig2e and h looks very different.

- In Fig 2c- I, definition of DN3a and DNb is done using CD27 as a marker, but Fig2c shows that CD27 expression is reduced in the cKO, so it does not seem appropriate it to use it to define these subsets. A better definition would be to compare icTCRbeta+ and icTCRbeta-neg DN3 and DN4 populations.

-It would be useful to have markers of proliferation and survival, death following pre-TCR signalling in the DN populations. Staining against cell surface CD71 expression on DN3a, DN3b, DN4icTCRB+, DN4icTCRB-neg, ISP would be useful (see Solanki et al 2020 <https://doi.org/10.1242/dev.192203>); cell surface CD2 expression on these subsets; AnnexinV staining on these subsets; cell surface CD69 staining on these subsets; ic cyclinB staining.

-For barcharts shown in Fig2.b-d, f, g staining (as well as MFI) should be shown. The very low MFI shown for example in f could be influenced by changes in expression of another marker (compensation on multicolor analysis) or FSC/SSC differences.

- Page 8 results the statement "suboptimal expression ofcontributes in part to defective early thymocyte maturation". This is incorrect. It isn't possible to know if these changes are causative, or a consequence of the defective development.

- Analysis of gammadelta cell development and populations: The authors should carry out a more thorough examination of gd cell development in the cKO, using markers of early development (CD25, CD73, CD24), and lineage (CD27, CD44, NK1.1, CD122, Vg-usage) to understand if the increase in gd cell numbers is the result of expansion of a particular subset or developmental stage. This is of interest, given that both ERK activation/TCR signal strength and Notch activation influence gd subset distribution (see Buus et al Nat Communications 8:1911). Additionally, cell death (annexinV staining) on gd subsets should be assessed.

- Fig.4i. Staining for phosphoERK not very convincing. Better to look at ERK activation downstream of TCR ligation (easy to do in peripheral T cells) by Western blot.

- Cell surface CD5 expression correlates to TCR signal strength and cell surface CD5MFI should be included for thymocyte and peripheral T-cells populations.

- Fig. 4d. Were CFSE experiments carried out on sorted cells or in FTOC? Were they given activation signals/growth factors? If in cell suspension the cells were not getting 'normal' signals from the thymus microenvironment. It seems strange to describe the gammadelta proliferation as 'sluggish'.

- Fig. 5e: why is percentage of DN1 cells so high? Show number of cells recovered in Fig. 5 as well as % of populations.

- Fig. 6b what happens to the number of gammadelta cells on treatment of cKO?

- Introduction of the transgenic OT-1 TCR: this data is not helpful because fully rearranged TCR is expressed too early.

- It would be useful to carry out competitive adoptive transfer experiments in which labelled coKO and WT progenitor cells are transferred into recipients

Reviewer #3 (Remarks to the Author):

The authors explore HBP pathway involvement in T cell ontogeny using the GFAT deficiencies and supplementation with GlcN to rescue. Data showing that PI3K/Akt signaling in the absence of GFAT1 is not sufficient to rescue T cell development suggesting that metabolic flux to specifically the HBP is necessary. The authors N- and O- linked and O-GlcNAcylation briefly in the introduction but do not mention reported functions in immune cells. For example, partial depletion of UDP-GlcNAc can induce ER stress and UPR by slowing N-glycosylation, and diminished surface expression of receptors by reducing Golgi processing. These are relevant for T cell ontogeny and activation. The author should be familiar with mechanistic work pioneered by other labs eg. PMID: 21629267, 25263124, 22288682.

The analysis in Fig 1-3 of signaling, metabolite profiling and gene expression are informative, but the role of HBP-dependent glycans in T cell ontogeny is left to speculation.

Page 8 bottom: CD98 is known to increase with ER and oxidative stress, and the metabolite data in Fig 3 is consistent with an amino acid and energy imbalance (eg. AMP/ATP). The feedback mechanism that upregulates CD98, CD147, CD25 and IL7R in GFAT1^{-/-} cell may be due to reduced N-glycosylation and thus ER stress which affects both gene expression and trafficking at the cells surface. TCR and the receptors mentioned above are N-glycosylated and modified in the Golgi with branched N-glycans that bind to galectins that regulate residency and dynamics at the cell surface. The authors do not explore an HBP dependent mechanisms.

Experiments with GlcN and DKG do not include a control with DKG alone. Why is this the only metabolite tested, - does pyruvate or Gln or glucose do the same? Does titration of GlcN show a dose response? As such it is not clear that GlcN is responsible for the partial rescue seen in Fig 5 or 6. GlcNAc increased UDP-GlcNAc levels - does it also rescue in culture?

Point-by-Point Reply to Reviewers (our replies are in **Purple Font**):

We would like to express our sincerest thanks to the reviewers for their prompt, careful, and overall positive review of our manuscript. We are delighted that the
revi

RESPONSE TO REVIEWERS' COMMENTS

Reply to Reviewer #1

This study examines the molecular mechanisms that control the development of abT-cells and gd-cells in the thymus. Given the multiple roles that these cells play in regulation and function of the immune response, studies of this kind are important, as understanding of intrathymic T-cell development remains incomplete. Specifically, the authors have examined the role of metabolism and the hexosamine biosynthesis pathway (HBP) in this process. While the importance of this pathway and the events that it controls are relatively well studied in other biological systems, there is little evidence for its important in the context of thymus biology. As such, the study is of interest and timely.

We sincerely appreciate the Reviewer's comment on the timeliness and importance of our study.

The authors show that use of mice carrying a T-cell specific deletion of Gfat1 (the rate limiting enzyme for de novo HBP synthesis) impairs ab but not gdT-cell development. This maps to alterations in the expression of multiple cell surface receptors known to play a role in T-cell development, including TCR, Notch and CXCR4. Finally, the authors show that dietary supplementation of T-cell deficient Gfat1 mice, and in vitro supplementation of FTOC, can overcome defective T-cell development. Overall, the study is of interest, and contains novel findings that open up further study of new pathways relevant to intrathymic T-cell development. There are several issues that need to be addressed:

1. In their study of T-cell specific Gfat1 deficient mice, the authors use p56lckCre to

drive deletion in thymocytes. While this is a fine model, the authors must use Cre only mice as controls, rather than the WT mice that are currently used (eg Fig1). This is because P56lckCre alone has previously been shown to result in thymus defects (Shi and Petrie Plos One 2012) which may well influence the degree of difference between control and experimental mice currently being used,

We appreciate the reviewer's suggestion. We did in fact consider this and have now included our results on the analysis of Lck-Cre^{+/-} to compare with the WT. As shown in **revised Fig 1b-j**, we did not find a significant difference in thymocyte number nor thymus weight in Lck-Cre^{+/-} mice. Since breeding *Lck-Cre^{-/-}/GFAT^{fl/fl} x Lck-Cre^{+/-}/GFAT^{fl/fl}* mice allows us to generate both *Lck-Cre^{-/-}/GFAT^{fl/fl}* (WT) and *Lck-Cre^{+/-}/GFAT^{fl/fl}* (KO) with true littermates as opposed to only generating age-matched mice as well as with no wasted animals due to their "wrong" genotype, we have used *Lck-Cre^{-/-}/GFAT^{fl/fl}* mice as our WT control throughout the experiments.

We would also like to point out that there was a further decrease in the rictor^{T-/-} and more importantly a more pronounced decrease occurred in the GFAT1^{T-/-}, further supporting that the phenotype we observed upon the loss of rictor or GFAT1 is not entirely due to the Cre protein levels as proposed by Shi and Petrie. (Note: the homozygous Lck-Cre^{+/+} that they also examined have more dramatic effects but our studies only employed Lck-Cre^{+/-}).

It is also worth noting that loss of PTEN (also driven by Lck-Cre^{+/-}) does not decrease cellularity, thus supporting that genetic deletion driven by Lck-Cre results in phenotypes that are independent from the effects of the Lck-Cre transgene or Cre expression. Our finding that the nutrient supplementation could rescue CD8 SP cell death in the GFAT1^{T-/-} (despite the presence of Lck-Cre^{+/-}) also strengthen the notion that nutrient metabolism impacts developmental phenotype of T cells in addition to genetic (and epigenetic) control. In summary, although Lck-Cre^{+/-} may have a slight effect on thymocyte cellularity likely due to Cre expression as reported by Shi and Petrie, we did not find dramatic effects in our system. Thus, our findings support a role for GFAT1 in early T cell development.

2. Can the authors include examples of FACS plots for CD27, Notch 1, CD98, GluT1, O-GlcNAcylation that are shown as MFI (eg Figures 2, 3)?

We have now included FACS plots. Please see revised figures:

Fig. 2c (CD27), Fig. 2d (Notch1), Fig. 2f (TCR β), Fig. 2g (icTCR β), Fig. 2i (CD25)

Fig. 3c (O-GlcNAc), Fig. 3d (CD98), Fig. 3e (GluT1)

new Fig. 5d (CD5), Fig. 5e (TCR β , $\gamma\delta$ TCR), **new Fig. 5f ($\gamma\delta$ TCR)**

Fig. 6b (TCR β , $\gamma\delta$ TCR), Fig. 6c (TCR β)

Fig. 7f (TCR β), Fig. 7k(TCR β)

Supplementary Fig. 2b (CXCR4), Supplementary Fig. 2c-d (TCR β)

Supplementary Fig. 2f-g (CD5), Supplementary Fig. 2h (CD147, CD98)

Supplementary Fig. 2i (CD127)

Supplementary Figure Fig. 6d (CD8), Supplementary Figure Fig. 6e (CD4),
Supplementary Figure Fig. 6f (CD44) and Supplementary Figure Fig. 6g (CD25).

In addition, we would like to point out that as indicated in the Methods section (line 608-609) our bar graphs measure “**median** fluorescence intensity” (MFI) of protein expression which is independent from cell numbers as opposed to other studies that report MFI as **mean** fluorescence intensity which is influenced by cell numbers.

3. Regarding comparative analysis of abT-cell development with gdT-cell development, it is important to note that in rescue experiments in vivo (fig 6) or in vitro (fig 5), the nature of the gdT-cells, and their developmental requirements, will be distinct. For example, in in vitro FTOC experiments: gdT-cells will be synchronised waves of cells expressing fetal invariant gdTCRs (eg dendritic epidermal T-cells), while in vivo experiments will analyse only adult gdT-cell development where this tissue restriction/limited TCR diversity is not evident. Do the authors have data on how fetal and adult gdT-cells are influenced by absence of Gfat1? Analysis of fetal/neonatal P56LckCreGfat1 floxed mice may go part way to addressing this.

This was a great suggestion. We now include analysis of $\gamma\delta$ -T cells from fetal/neonates and adult mice. As shown in **new Fig. 4d-f**, we found that the generation of $\gamma\delta$ -TCR diversity is compromised despite increased $\gamma\delta$ -T cell numbers during GFAT1 deficiency (**see Results pages 13-14, lines 291-311**). These findings further support the notion that the *de novo* synthesis of hexosamines (by GFAT1) provide sufficient hexosamines that are required for proper TCR synthesis and diversification. In other words, our findings underscore the “supply and demand” concept such that GFAT1 ensures availability of sufficient hexosamines during developmental stages when hexosamine demands are high and cannot be met by salvage mechanisms. Hence, although development of $\alpha\beta$ -T cells are dependent on dn-HBP due to the generation of a highly diverse $\alpha\beta$ -TCR repertoire, we found that even $\gamma\delta$ -T cells that have relatively limited diversity in $\gamma\delta$ -TCR still rely to some extent on dn-HBP. We have modified the discussion and the abstract to reflect these new findings (**see Discussion page 23, lines 526-538, and Abstract page 2, lines 8-9**). Our main conclusion is not changed by these new findings but instead further support how sufficient hexosamines are required to meet the demand for this metabolite for proper synthesis of receptors involved during early T cell development. Importantly, these new findings open up new area of investigation as to precisely how hexosamines or other metabolites are required during TCR diversification.

Reply to Reviewer #2

This is a very interesting and novel study that shows the importance of GFAT1 and

hexosamine biosynthesis in T-cell development. This is timely and of general interest to the community.

We sincerely appreciate the Reviewer's comment on the novelty and timeliness of our studies.

However, I believe the manuscript requires major revision to be ready for publication, particularly in relation to the analysis of thymocyte subsets and T-cell development

-throughout the figures, the term "relative cell number (%)" is used to label bar chart y-axis which in fact show percentages (not relative cell number). This is confusing and the focus on comparison of the percentage or proportion of a thymocyte population rather than the number of cells in the population is confusing and misleading.

We apologize for this confusion. In addition to the total cell number, we included the relative cell numbers in order to compare the proportion of each thymocyte subset within a given population. This is particularly important given the profound decrease in total cell numbers in GFAT1^{T-/-}. The "relative cell number" allows us to distinguish a potential developmental block in a particular subset as this would be indicated by an increase in the percentage relative to the WT subset. In any case, to avoid confusion we relabeled the plots as "subset proportion (%)" and clarified this labeling in the corresponding Figure Legend. Please see **revised Figures 1g, 2a, 4b, 4e, 6a, 6d, 6e, 7b, 7d, 7h**).

The analysis of number of cells clearly shows that the number of DN cells is normal, but a partial developmental arrest occurs from ISP onwards (Fig. 1b, h, j). Therefore, the fact that the percentage of DN cells is increased is not very relevant, and is confusing.

We thank the reviewer for bringing up this point and we therefore further examined this. We have analyzed more WT (both GFAT1^{fl/fl} and Lck-Cre^{+/-}) to increase sample numbers (from n=6 to n=8 in GFAT1^{fl/fl} and n=4 in Lck-Cre^{+/-}). Statistical analysis now shows a significant increase in the DN cell number in GFAT1^{T-/-} (**revised Fig 1 h**). Therefore, both proportion (Fig. 1g) and number of DN cells (Fig. 1h) in the GFAT1^{T-/-} are significantly higher than WT.

-analysis of DN subsets: the manuscript states that the DN subsets are gated as Lin⁻neg and staining includes CD4, CD8, CD3e, NK1.1, B220, Ter119, Gr1. Presumably this means that CD3e⁺ cells have been gated out of the analyses of the data shown in Fig2 (excluding NKT-cells, gammadelta cells); however, if CD3 has been gated out, then cell surface TCRbeta⁺ cells should also have been excluded, but Fig. 2e, h, show around 30% WT DN4 cells express cell surface TCRbeta. This does not seem right: at this stage of abT-cell development expression of the pre-TCR complex (TCRbeta with the chains of CD3) is very low but icTCRbeta expression is easy to detect.

The reviewer brings up an important point that remains to be fully explored in the field: does the pre-TCR have the same CD3 configuration/quantity as $\alpha\beta$ TCR? Nevertheless, for our studies we employed the conventional method of identifying DN cells and excluded **high** CD3 ϵ ⁺ to avoid counting mature cells that have potentially downregulated CD4 and/or CD8 (eg negative selection of DP cells) and would appear as DN cells. We have used a low amount (1/400 dilution) of anti-CD3 ϵ to avoid exclusion of low CD3 ϵ ⁺ cells (DN4TCR β ⁺).

Additionally the surface TCRbeta staining on WT between Fig2e and h looks very different.

Figure 2e plots Notch1 vs TCR β , thus staining was done on both surface proteins. In contrast, Figure 2h plots intracellular TCR β vs surface TCR β . The difference in staining and some slight variability between independent experiments could account for the distinct profiles between 2e and 2h. Nevertheless, both figures reveal that among the proportion of DN3b cells that express higher TCR β , there is roughly 10-fold decrease (8.54 vs 0.86) in the proportion of cells expressing Notch1 (2e) and a 3-fold decrease (8.43 vs 2.92) in proportion of cells expressing ic-TCR β (2h) in the absence of GFAT1. In DN4 cells that express higher TCR β , there is about 25-fold decrease (4.28 vs 0.17) in proportion of cells expressing Notch1 (2e) and a 9-fold decrease (35.9 vs 4.12) in proportion of cells expressing ic-TCR β (2h) in the absence of GFAT1.

- In Fig 2c- I, definition of DN3a and DNb is done using CD27 as a marker, but Fig2c shows that CD27 expression is reduced in the cKO, so it does not seem appropriate it to use it to define these subsets.

The markers we use to designate the defined subsets are indeed based on wild type and the convention used in the field. Hence, the levels of expression of cell surface proteins in the absence of GFAT1 are relative to the WT. We would also like to point out that our thymocytes were stained for several cell surface markers (as indicated in the Figure legend, see Fig. 2 legend) to define the different subsets. One of the key findings in our studies is that the majority of key receptors/cell surface proteins including $\alpha\beta$ TCR is diminished in the absence of GFAT1. Nevertheless, there are some exceptions such as CD25, which is highly upregulated in the DN cells of GFAT1^{T-/-} (Fig. 2i). In summary, the cell surface markers we use allow us to define the subset based on WT levels. Whether there are cell surface proteins that remain unaltered in the absence of GFAT1 (in DN3a vs DN3b stages or other stages) could be addressed in future investigation.

A better definition would be to compare icTCRbeta⁺ and icTCRbeta⁻ DN3 and DN4 populations.

We appreciate these suggestions and agree that these experiments could be an alternative to gain additional insights on changes at the DN3 stage. As shown below we have gated the same WT and GFAT1^{T-/-} DN3 samples for a) CD27 vs FSC or b) icTCRβ vs sTCRβ to determine DN3a (CD27⁻/FSC^{low} vs icTCRβ⁻/sTCRβ⁻) and DN3b (CD27⁺/FSC^{high} vs icTCRβ⁺/sTCRβ^{low}) subsets. Both approaches revealed fairly similar DN3a and DN3b proportions for WT thymocytes, thus, validating the use of either approach to measure DN3a and DN3b subsets. Our finding with CD27 staining was that the loss of GFAT1 reduces the differentiation of DN3a thymocytes into DN3b subsets as compared to WT counterparts (a). The icTCRβ gating confirms these findings, albeit with a less pronounced difference as compared to WT cells (b). This was probably in part due to a higher number of GFAT1-deficient DN3 cells that accumulate mis-processed icTCRβ and thus using this gating method would fairly overestimate the proportion of DN3b in GFAT1^{T-/-} mice. Therefore, we decided to keep the traditional CD27 staining to assess DN3a vs DN3b subsets.

-It would be useful to have markers of proliferation and survival, death following pre-TCR signalling in the DN populations.

Figures 4g shows analysis of DN proliferation using CFSE whereas 4h-i shows analysis of live cells upon labeling with Annexin V. Since we did not stimulate with anti-CD3 *ex vivo*, our analyses are unbiased by the expression levels of the pre-TCR or γδTCR and its consequence on signaling shifts. In this revision, we have now also analyzed proliferation and survival of other subsets. In addition to analysis of proliferation of DN thymocytes (Fig. 4g) and CD8-ISP (Supplementary Fig. 4c), we now also include proliferation analysis in DP cells (**new Supplementary Fig. 4d**). These new findings further support reduced proliferation in these different subsets in the absence of GFAT1. In addition to cell survival analysis of TCRβ⁺ DN4 and

$\gamma\delta$ TCR⁺ DN cells (Fig. 4h-i), we now also include cell survival analysis of CD8-ISP (**new Supplementary Fig. 4e**) and DP cells (**new Supplementary Fig. 4f**). These findings support that GFAT1 is essential for optimal cell survival.

Staining against cell surface CD71 expression on DN3a, DN3b, DN4icTCRB+, DN4icTCRB-neg, ISP would be useful (see Solanki et al 2020 <https://doi.org/10.1242/dev.192203>); cell surface CD2 expression on these subsets; AnnexinV staining on these subsets; cell surface CD69 staining on these subsets; ic cyclinB staining.

We appreciate these suggestions and agree that these experiments will provide additional insights on changes in expression level of other receptors and cell signaling/proliferation. However, due to very limited amounts of thymocytes acquired from GFAT1^{T-/-} mice, we had to prioritize and therefore decided to conduct the most crucial experiments suggested by all three reviewers and highlighted by the editor. We apologize that we were not able to address these suggested experiments from this reviewer. We hope that the reviewer agrees that we have provided substantial new data to support the main conclusion of our studies.

-For barcharts shown in Fig2.b-d, f, g staining (as well as MFI) should be shown.

As suggested, we have now included the data for staining, in addition to the bar charts:

Fig. 2c (CD27), Fig. 2d (Notch1), Fig. 2f (TCR β), Fig. 2g (icTCR β), Fig. 2i (CD25)
Fig. 3C (O-GlcNAc), Fig. 3d (CD98), Fig. 3e (GluT1)

new Fig. 5d (CD5), Fig. 5e (TCR β , $\gamma\delta$ TCR), **new Fig. 5f ($\gamma\delta$ TCR)**

Fig. 6b (TCR β , $\gamma\delta$ TCR), Fig. 6c (TCR β)

Fig. 7f (TCR β), Fig. 7k(TCR β)

Supplementary Fig. 2b (CXCR4), Supplementary Fig. 2c-d (TCR β)

Supplementary Fig. 2f-g (CD5), Supplementary Fig. 2h (CD147, CD98)

Supplementary Figure Figure Fig. 2i (CD127).

Fig. 2b plots the total cell number, which was not measured by flow cytometry, hence we do not have staining for that figure. As indicated in the Methods, total cell number was calculated by multiplying the cell counts by the relative number of subsets in a pool of cells.

In addition, we would like to point out that as indicated in the Methods section (line 608-609) our bar graphs measure “**median** fluorescence intensity” (MFI) of protein expression which is independent from cell number as opposed to other studies that report MFI as **mean** fluorescence intensity which is influenced by cell numbers.

The very low MFI shown for example in f could be influenced by changes in expression of another marker (compensation on multicolor analysis) or FSC/SSC differences.

We appreciate the reviewer's concern about fluorescence bleaching from channel to channel and would like to point out that compensation parameters were reset prior to every acquisition experiment. Moreover FSC/SSC gating was kept constant throughout every experiment. However, we do believe that the low levels of TCR β surface expression shown in Fig. 2f reflect the low level of pre-TCR expression as compared to the much higher levels of $\alpha\beta$ TCR observed on CD4- and CD8-SP cells as shown in Supplementary Fig. 2C. If DN TCR β MFI was as high as for example DP, this would probably indicate a high rate of more mature thymocytes that have downregulated their CD4, CD8 coreceptors and thus wrongly gated as DN cells. We do exclude those potential false DN cells from our DN gating by staining with a low amount of CD3 ϵ (1/400).

- Page 8 results the statement "suboptimal expression ofcontributes in part to defective early thymocyte maturation". This is incorrect. It isn't possible to know if these changes are causative, or a consequence of the defective development.

We have now rephrased this sentence (**page 8, lines 160-162**) as follows: "The defective $\alpha\beta$ -T cell development in the absence of GFAT1 is accompanied by suboptimal expression of CD27, Notch1, CXCR4 and the pre-TCR β -chain at the DN3b stage."

- Analysis of gammadelta cell development and populations: The authors should carry out a more thorough examination of gd cell development in the cKO, using markers of early development (CD25, CD73, CD24), and lineage (CD27, CD44, NK1.1, CD122, Vg-usage) to understand if the increase in gd cell numbers is the result of expansion of a particular subset or developmental stage.

We thank the reviewer (as well as reviewer 1) for this great suggestion. We now include analysis of $\gamma\delta$ -T cells from fetal/neonates and adult mice. As shown in **new Fig. 4d-f** we found that the generation of $\gamma\delta$ -TCR diversity is compromised despite increased $\gamma\delta$ -T cell numbers during GFAT1 deficiency. These findings further support the notion that the *de novo* synthesis of hexosamines (by GFAT1) provide sufficient hexosamines that are required for proper TCR synthesis and diversification. In other words, our findings underscore the "supply and demand" concept such that GFAT1 ensures availability of sufficient hexosamines during developmental stages when hexosamine demands are high and cannot be met by salvage mechanisms. Hence, although development of $\alpha\beta$ -T cells are dependent on dn-HBP due to the generation of a highly diverse $\alpha\beta$ -TCR repertoire, we found that even $\gamma\delta$ -T cells that have relatively limited diversity in $\gamma\delta$ -TCR still rely to some extent on dn-HBP. We have modified the discussion (**see page 23, lines 526-544**) and the abstract (**page 2, lines 8-9**) to reflect these new findings. Our main conclusion is not changed by these new findings but instead open up new area of investigation as to precisely how hexosamines are required during TCR diversification.

This is of interest, given that both ERK activation/TCR signal strength and Notch activation influence gd subset distribution (see Buus et al Nat Communications 8:1911). Additionally, cell death (annexinV staining) on gd subsets should be assessed.

The analysis of cell death on $\gamma\delta$ T cell subsets is presented on Fig. 4i (formerly Fig. 4f).

- Fig.4i. Staining for phosphoERK not very convincing. Better to look at ERK activation downstream of TCR ligation (easy to do in peripheral T cells) by Western blot.

We would like to clarify that we conducted the analysis of ERK phosphorylation in Fig. 5c (formerly Fig. 4i) using immunostaining/flow cytometry since we wanted to discriminate mature (SP) from immature (DN4) cells from lineage committed (gamma delta vs alpha beta) cells. An alternative would be to sort DN cells. However, due to limited amount of thymocytes that can be recovered from GFAT1-deficient cells, it is technically challenging to obtain sufficient amounts for sorting followed by Western blotting. Hence, although we could perform immunoblotting in unsorted peripheral T cells, this method would also present other caveats/limitations that will not directly answer our question as to how ERK signaling is affected in the DN subsets that are relevant for gamma delta T cell differentiation. Nevertheless, we followed the suggestion of the reviewer using a modified approach as follows. We used OT-1 RagKO background for these analysis in order to obtain sufficient thymocytes in the absence of GFAT1 and the RagKO background enabled exclusion of B cells from the spleen. As demonstrated by these results, ERK phosphorylation was dramatically increased upon α CD3 or PMA stimulation in both thymocytes and peripheral T cells. However, its phosphorylation was comparable in the WT and GFAT1-deficient cells. A possible interpretation is that sufficient expression of OT-1 facilitated strong ERK signaling even in the absence of GFAT1. Since DP cells represent the majority of total thymocytes (in WT and GFAT1^{T-/-}), these results likely reflect intracellular signaling of *ex vivo*-stimulated DP cells. These findings indicate that in the absence of GFAT1, intrinsic ERK signaling is not defective when maximal signals are provided *ex vivo* (eg by α CD3 or PMA stimulation) using total thymocytes or peripheral T cells.

The levels of ERK phosphorylation upon TCR/CD3 or PMA stimulation in total thymocytes or splenic T cells are comparable in WT vs GFAT1-deficient cells. Thymocytes or splenocytes harvested from OT-1 RagKO background (to exclude B cells in the spleen) WT (GFAT1^{fl/fl}) or GFAT1^{-/-} mice were incubated *ex vivo* in complete media with or without α CD3 antibody (30 μ g/mL) or PMA (100 nM) for the indicated times. Cells were lysed in RIPA lysis buffer, then lysates were fractionated SDS-PAGE and subjected to immunoblotting using the indicated antibodies.

- Cell surface CD5 expression correlates to TCR signal strength and cell surface CD5MFI should be included for thymocyte and peripheral T-cells populations.

We have now analyzed cell surface CD5 expression. Our results indicate that its expression was slightly increased in DN subsets (both TCR β - and $\gamma\delta$ -TCR-expressing cells) of GFAT1-deficient thymocytes (**new Fig. 5d**), consistent with the increased ERK phosphorylation in these subsets (Fig. 5c [formerly Fig.4i]) (**see Results page 16, lines 369-371**). These findings are thus in line with the notion that increased TCR/ERK signaling skews development towards $\gamma\delta$ lineage. In contrast, we found that starting at the DP stage and onward, including peripheral $\alpha\beta$ -T cells, CD5 has decreased expression in the GFAT1-deficient cells (**new supplementary Fig. 2f-g**). The relevance of this reduced CD5 expression to TCR signal strength is unclear at the moment. Therefore, we chose to exclude these findings from the main study. Whether CD5 acts as a positive or negative regulator of TCR signaling in this context would also need to be further examined (Burgueno-Bucio et al., 2019). Nevertheless, given that CD5 undergoes *N*- and *O*-glycosylation (Pospisil et al., 2009), the decreased overall expression of CD5 in thymocytes is consistent with possible defects in glycosylation due to GFAT1 deficiency.

- Fig. 4d. Were CFSE experiments carried out on sorted cells or in FTOC?

For Fig. 4g (formerly Fig. 4d), the CFSE experiments were gated on live thymocytes from 5 wk old mice and then gated on the respective DN4 or $\gamma\delta$ DN subsets. We clarify this now in the Figure legend as follows:

Figure 4:

g. Thymocytes from 5 wk old mice were labeled with CFSE and cultured *ex vivo* for 24h or 48 hr in complete DMEM media. Cells were harvested and stained for CD4, CD8 α , CD3 ϵ , NK1.1, B220, Ter119, Gr1, CD25, CD44, TCR β or $\gamma\delta$ TCR, followed by flow cytometry. Gating was set on live thymocytes and either TCR β ⁺-DN4 (upper panel) or $\gamma\delta$ TCR⁺ DN cells (lower panel).

Were they given activation signals/growth factors?

If in cell suspension the cells were not getting 'normal' signals from the thymus microenvironment.

In Fig. 4g (formerly Fig. 4d), the analysis was done at basal levels, ie, no additional activation signals other than the culture media: DMEM with 10%FCS, 25 mM glucose, 1% glutamine, 1% pen/strep. Indeed, nutrients and growth factors from the media would influence growth and proliferation. The purpose of this experiment was to analyze whether there is a basal proliferation defect in the absence of GFAT1. The results in Fig. 4g reveal that there is decreased basal proliferation in the absence of GFAT1 in both TCR β + DN4 and $\gamma\delta$ TCR+ DN thymocytes. However, since the recovery (prior to *ex vivo* culture) of live cells from GFAT1^{T-/-} is profoundly diminished (by 50%) compared to WT (Fig. 4h), these findings indicate that GFAT1 is essential for the survival and/or proliferation of $\alpha\beta$ -lineage cells in the thymic environment (*in vivo*). We have clarified this point in the Results section (**page 14, lines 320-321**) as follows:

These findings indicate that GFAT1 is essential for the survival and/or proliferation of $\alpha\beta$ -lineage cells in the thymic environment *in vivo*.

It seems strange to describe the gammadelta proliferation as 'sluggish'.

We have replaced this with "slower", see Discussion (**page 23, line 527**).

- Fig. 5e: why is percentage of DN1 cells so high?

This analysis was done on FTOCs that could have higher DN1 proportion. A previous study that compared proliferation and differentiation in FTOC vs thymocytes *in vivo* has also reported higher DN and overall less proliferation in FTOC compared to thymocytes *in vivo* (Zhang et al, 2007).

Show number of cells recovered in Fig. 5 as well as % of populations.

Unfortunately, we do not have the cell numbers after supplementation in Fig. 6 (formerly Fig. 5). Hence, we did not include recovered cell numbers for the supplemented FTOCs. Due to limited availability of the GFAT1-deficient mice and time limitation, we apologize to the reviewer as we were unable to repeat these set of experiments to provide data for this request.

- Fig. 6b what happens to the number of gammadelta cells on treatment of cKO?

In **new Fig. 7b** (formerly Fig. 6b) we demonstrate that the proportion of $\gamma\delta$ -T cells was reduced to WT levels after GlcN/DKG supplementation of GFAT1^{T-/-} mice. However, we did not see a significant reduction of $\gamma\delta$ -T cell number (**new Supplementary Fig. 7a**).

- Introduction of the transgenic OT-1 TCR: this data is not helpful because fully rearranged TCR is expressed too early.

This point is well taken that the full transgenic TCR expression occurs at DN3-DN4 as opposed to DP. Surprisingly, despite putatively earlier (and most likely stronger) signaling from the TCR, this is not sufficient to drive more DN or ISP cells towards maturation (Supplementary Fig. 6a-b). Thus, we chose to include these data to support the idea that although signals from the TCR are critical for early T cell development, they are not sufficient if key metabolic pathways (eg *de novo* HBP) are defective. In addition, for this revision, we have also conducted glycomics analysis as well as lectin-pulldown experiments of TCR β (in response to Rev. 3 critique) using the OT-1 TCR background in WT vs GFAT1^{T-/-} thymocytes (**see new Fig. 3b and new Supplementary Fig. 3a**). Using this genetic background validated defective N-glycosylation in the absence of GFAT1. Hence, we hope that the reviewer will agree that the OT-1 TCR mice have been useful for our purposes in addressing the role of GFAT1 during development.

- It would be useful to carry out competitive adoptive transfer experiments in which labelled coKO and WT progenitor cells are transferred into recipients.

This is indeed a good experiment to further support the role of GFAT1 in early T cell development. However, due to the limited time and resources at this point, we are unable to perform these suggested experiments. We hope that the additional analysis that we have provided in response to the key issues raised by this reviewer and the other reviewers suffice to strengthen our conclusion.

Reply to Reviewer #3

We thank the reviewer for careful review of our work and for the relevant suggestions.

The authors explore HPB pathway involvement in T cell ontogeny using the GFAT deficiencies and supplementation with GlcN to rescue. Data showing that PI3K/Akt signaling in the absence of GFAT1 is not sufficient to rescue T cell development suggesting that metabolic flux to specifically the HBP is necessary. The authors N- and O- linked and O-GlcNAcylation briefly in the introduction but do not mention reported functions in immune cells. For example, partial depletion of UDP-GlcNAc can induce ER stress and UPR by slowing N-glycosylation, and diminished surface expression of receptors by reducing Golgi processing. These are relevant for -T cell ontogeny and activation. The author should be familiar with mechanistic work pioneered by other labs eg. PMID: 21629267, 25263124, 22288682.

We appreciate this suggestion from the reviewer to further discuss the role of *N*- and *O*-linked glycosylation in immune cells. Importantly, we highlight the effects of UDP-GlcNAc depletion in inducing ER stress and UPR. We have now added some key references to further support the role of proper *N*-glycosylation during T cell development and activation. Since our studies have limited data on *O*-GlcNAcylation, we did not further expand this discussion other than what has been already mentioned. However, we added a concluding statement in that paragraph to emphasize that the effects of GFAT1 deficiency on specific *N*- and *O*-GlcNAcylation during T cell development would need to be investigated in future studies. **Please see revised Discussion, pages 21-22, lines 498-509.**

The analysis in Fig 1-3 of signaling, metabolite profiling and gene expression are informative, but the role of HBP-dependent glycans in T cell ontogeny is left to speculation.

This is indeed an important point that needs to be further addressed in future studies. In this revision, we conducted some new studies to begin to address this issue. To further support that the defects in GFAT1-deficient thymocytes are due to impaired HBP, we have now conducted *N*-glycomics analysis to compare WT vs GFAT1-deficient thymocytes. Since the GFAT1^{T-/-} mice have profoundly decreased thymocyte numbers, we utilized thymocytes from OT-1 transgenic mice (GFAT1^{T+/+} vs GFAT1^{T-/-}) in order to obtain sufficient cell numbers for mass spec analysis of the *N*-glycome. We now present in the **new Supplementary Fig. 3a** that in the GFAT1-deficient thymocytes, there were higher oligomannose-type glycans and lower complex/hybrid-type glycans in comparison to the GFAT1^{T+/+}. These findings support that in absence of GFAT1, the reduced UDP-GlcNAc levels (Fig. 3a) could lead to defective *N*-glycans, in addition to the diminished *O*-GlcNAcylation of total proteins (Fig. 3c).

To substantiate that there is defective TCR glycosylation, we evaluated TCR β expression levels and binding to specific lectins. Defective glycosylation leads to misfolding and consequently protein degradation. We therefore treated OT-1/WT and OT-1/GFAT1^{T-/-} thymocytes with the proteasome inhibitor MG132 to prevent degradation of improperly folded TCR. As shown in **new Fig. 3b**, we found that whereas OT-1/GFAT1^{T-/-} cells have decreased levels of TCR β , the amounts were restored to OT-1/WT levels upon MG132 treatment. Furthermore, upon lectin pull-down of total extracts using GNL (*Galanthus nivalis* lectin), there was increased binding of TCR β from MG132-treated as compared to non-treated GFAT1-deficient cells. In contrast, MG132 did not alter the amount of TCR β pulled-down by the lectin SNA from WT or GFAT1-deficient thymocyte extracts. Since GNL binds to α -1,3 mannose that are exposed in the early steps of *N*-glycan remodeling while SNA binds to terminal sialic acid residues, our results suggest that early glycan misprocessing of TCR β , leading to its premature degradation, occur in the absence of GFAT1. These findings indicate defective TCR β glycosylation in the absence of

GFAT1, supporting that impaired HBP leads to developmental defects in GFAT1-deficient thymocytes. Please see **Results section pages 9-10, lines 187-215, and Discussion pages 21-22, lines 498-509.**

Page 8 bottom: CD98 is known to increase with ER and oxidative stress, and the metabolite data in Fig 3 is consistent with an amino acid and energy imbalance (eg. AMP/ATP). The feedback mechanism that upregulates CD98, CD147, CD25 and IL7R in GFAT1^{-/-} cell may be due to reduced N-glycosylation and thus ER stress which affects both gene expression and trafficking at the cells surface. TCR and the receptors mentioned above are N-glycosylated and modified in the Golgi with branched N-glycans that bind to galectins that regulate residency and dynamics at the cell surface. The authors do not explore an HBP dependent mechanisms.

We appreciate these comments from the reviewer. We improved the discussion to include these interesting points, particularly how defective HBP could affect N-glycosylation and consequently trigger ER stress and metabolic reprogramming (see pages 21-22, lines 498-509). Furthermore, as indicated above, we have now examined N-glycosylation defects during GFAT1 deficiency by examining the N-glycome and conducting lectin pull-downs of TCR β (see **new Supplementary Fig. 3a** and **new Fig. 3b**). More specific N-glycosylation defects of other key cell surface receptors certainly warrant further investigation. The feedback mechanisms as well as metabolic reprogramming that could occur as a consequence of reduced UDP-GlcNAc/N-glycosylation, leading to changes in gene expression, trafficking and retention at the cell surface are all very interesting points that also deserve further interrogation but are beyond the scope of our current studies.

Experiments with GlcN and DKG do not include a control with DKG alone.

We were limited with FTOC availability and the few sole DKG treatments that we performed, we were unable to obtain sufficient replicates to do statistical analysis. When we perform FTOC experiments, we are blinded by the genotype since we need to perform the experiments asap upon dissection of the fetus. Hence, it is only after we analyze an aliquot of the tissue that we are informed of the genotype of our sample.

For the *in vivo* dietary supplementation, we only conducted GlcN/DKG supplementation since our results from the FTOC revealed enhanced rescue using this combined treatment. Since we were limited with the availability of GFAT1^{T^{-/-}} mice, we conducted our experiments only with these combined nutrients at a single time point (1 month). Nevertheless, to further interrogate the effects of GlcN/DKG treatment, we have now performed analysis of total metabolites of thymocytes after

in vivo GlcN/DKG treatment. As shown in the **new Fig. 7g**, there was a slight but insignificant increase in UDP-GlcNAc upon GlcN/DKG treatment of GFAT1^{T-/-} mice.

Why is this the only metabolite tested, - does pyruvate or Gln or glucose do the same? Does titration of GlcN show a dose response? As such it is not clear that GlcN is responsible for the partial rescue seen in Fig 5 or 6. GlcNAc increased UDP-GlcNAc levels - does it also rescue in culture?

Our goal for the dietary supplementation experiment is to determine whether bypassing *de novo* hexosamine biosynthesis (via a salvage metabolite, glucosamine [GlcN]) could rescue some or all of the defects in early thymocyte development in GFAT1^{T-/-}. We picked GlcN over *N*-acetylglucosamine (GlcNAc), which is the other salvage HB metabolite since GlcN is known to be efficiently transported by glucose transporters and is a widely used nutraceutical. In contrast, GlcNAc lacks a cell surface transporter and requires high doses for macropinocytotic entry (Lee et al., 2019). Addition of other metabolites may or may not rescue some or all of the phenotype but would require more extensive investigation to explain their effects. Hence, the effects of other metabolites on GFAT1^{T-/-} in comparison to glucosamine supplementation would indeed be worthy to examine in the future but is beyond the scope of the current studies. We did not conduct GlcN dose response studies due to limitation with the availability of GFAT1^{T-/-} mice. Variable concentrations for glucosamine feeding in mice have been used in the literature ranging from 1 mg to 10g (Tannock et al., 2006; Weimer et al., 2014; Yuan et al., 2021). Most of these studies have conducted feeding from 4-20 wks. Our studies have used 20 mg for 4 wk. It would indeed be important to address in the future whether varying the dose and timing would improve the effects of glucosamine.

Nevertheless, to further support that our supplementation strategy rescues metabolic defects in GFAT1-deficient thymocytes, we have now included metabolomics studies on thymocytes to compare non-supplemented vs GlcN/DKG-supplemented mice. As shown in **new Fig. 7g** GlcN/DKG supplementation slightly but significantly enhanced the levels of UDP-GlcNAc in GFAT1^{T-/-}. Unexpectedly, we found that the defective levels of nucleotides in GFAT1^{T-/-} were fully restored to WT levels upon GlcN/DKG supplementation (**new Supplementary Fig. 7c**). We further discuss the implications of these findings in the **Discussion page 25, lines 576-585**.

REFERENCES

Burgueno-Bucio, E., Mier-Aguilar, C.A., and Soldevila, G. (2019). The multiple faces of CD5. *Journal of leukocyte biology* 105, 891-904. 10.1002/JLB.MR0618-226R.

Lee, S.U., Li, C.F., Mortales, C.L., Pawling, J., Dennis, J.W., Grigorian, A., and Demetriou, M. (2019). Increasing cell permeability of *N*-acetylglucosamine via 6-acetylation enhances

capacity to suppress T-helper 1 (TH1)/TH17 responses and autoimmunity. *PLoS One* 14, e0214253. 10.1371/journal.pone.0214253.

Pospisil, R., Kabat, J., and Mage, R.G. (2009). Characterization of rabbit CD5 isoforms. *Mol Immunol* 46, 2456-2464. 10.1016/j.molimm.2009.05.026.

Tannock, L.R., Kirk, E.A., King, V.L., LeBoeuf, R., Wight, T.N., and Chait, A. (2006). Glucosamine supplementation accelerates early but not late atherosclerosis in LDL receptor-deficient mice. *The Journal of nutrition* 136, 2856-2861. 10.1093/jn/136.11.2856.

Weimer, S., Priebes, J., Kuhlow, D., Groth, M., Priebe, S., Mansfeld, J., Merry, T.L., Dubuis, S., Laube, B., Pfeiffer, A.F., et al. (2014). D-Glucosamine supplementation extends life span of nematodes and of ageing mice. *Nature communications* 5, 3563. 10.1038/ncomms4563.

Yuan, X., Zheng, J., Ren, L., Jiao, S., Feng, C., Du, Y., and Liu, H. (2021). Glucosamine Ameliorates Symptoms of High-Fat Diet-Fed Mice by Reversing Imbalanced Gut Microbiota. *Front Pharmacol* 12, 694107. 10.3389/fphar.2021.694107.

Zhang, J., Gong, Y., Shao, X., Zhang, R., Xu, W., Chu, Y., Wang, Y., and Xiong, S. (2007). Asynchronism of thymocyte development in vivo and in vitro. *DNA Cell Biol* 26, 19-27. 10.1089/dna.2006.0525.

REVIEWERS' COMMENTS

Reviewer #1 (expert in T cell development, thymic microenvironment, T cell lineage commitment):

The authors have clarified the points in my initial review, and provided additional data through new experiments that substantially strengthens their study and conclusions.

Reviewer #2 (expert in T cell development, signaling and function, lineage fate):

This is a thorough and interesting study. In my view the authors have addressed the reviewers' comments sufficiently to merit publication without further revisions.

Reviewer #3 (expert in hexosamine synthesis, O-GlcNAc modification, immunometabolism and cancer):

The revised version of this manuscript includes new data on glycosylation and surface expression of T receptors, as I had suggested. Importantly, the text has been modified to include a discussion of the results with reference prior literature on down-stream effectors of HBP. More could always be done, but the authors have done an admirable job of responding. I recommend publication without further revisions.